# Learning the Optimal Policy for Balancing Short-Term and Long-Term Rewards

**Qinwei Yang**[1], **Xueqing Liu**[1], **Yan Zeng**[1], **Ruocheng Guo**[2], **Yang Liu**[3], **Peng Wu**[1]*
[1]Beijing Technology and Business University    [2]ByteDance Research    [3]UC Santa Cruz

## Abstract

Learning the optimal policy to balance multiple short-term and long-term rewards has extensive applications across various domains. Yet, there is a noticeable scarcity of research addressing policy learning strategies in this context. In this paper, we aim to learn the optimal policy capable of effectively balancing multiple short-term and long-term rewards, especially in scenarios where the long-term outcomes are often missing due to data collection challenges over extended periods. Towards this goal, the conventional linear weighting method, which aggregates multiple rewards into a single surrogate reward through weighted summation, can only achieve sub-optimal policies when multiple rewards are related. Motivated by this, we propose a novel decomposition-based policy learning (DPPL) method that converts the whole problem into subproblems. The DPPL method is capable of obtaining optimal policies even when multiple rewards are interrelated. Nevertheless, the DPPL method requires a set of preference vectors specified in advance, posing challenges in practical applications where selecting suitable preferences is non-trivial. To mitigate this, we further theoretically transform the optimization problem in DPPL into an $\varepsilon$-constraint problem, where $\varepsilon$ represents the minimum acceptable levels of other rewards while maximizing one reward. This transformation provides intuitive into the selection of preference vectors. Extensive experiments are conducted on the proposed method and the results validate the effectiveness of the method.

## 1 Introduction

Learning an optimal policy for balancing multiple short-term and long-term rewards holds extensive applications across various domains. For instance, content providers can optimize recommendations to avoid short-term clickbait strategies, ensuring sustained user engagement and revenue growth [1]. IT companies can design web pages catering to immediate user preferences while enhancing long-term engagement and satisfaction [2]. Economists explore the effects of early childhood interventions on lifetime earnings, seeking optimal policies (e.g., class size) maximizing short-term test scores and long-term earnings simultaneously [3]. Policymakers can improve job training program design, considering both immediate income impacts and subsequent employment status improvements [4, 5]. Medical practitioners can refine drug prescriptions, considering short-term alleviation and long-term outcomes in chronic diseases like Alzheimer's and AIDS [6]. Marketing professionals can optimize incentive strategies to positively influence customer behavior in both short and long terms [7].

Despite the importance of balancing multiple short-term and long-term rewards, policy learning methods in this area remain largely unexplored. Recent literature [8] employs a linear weighting method to achieve this goal. It combines multiple rewards into a single surrogate reward by weighted summation, which is optimized to learn the optimal policy. However, this strategy has several limitations. First, it can only find optimal solutions in convex regions of objective space and cannot obtain the optimal solutions in non-convex regions [9]. Second, it achieves the optimal solution only

---

*Corresponding author: pengwu@btbu.edu.cn.

when the rewards are independent of each other. When some of the rewards are interrelated, it can only achieve sub-optimal solutions [10]. Consequently, although the linear weighting method is easy to implement, the optimality of its solution cannot be guaranteed when balancing multiple objectives.

In this article, we propose a principled policy learning approach for balancing multiple long-term and short-term rewards (objectives). Specifically, we first formulate it as a multiple-objective problem (MOP) and aim to seek the Pareto optimal solutions (policies). A solution is Pareto optimal if improving one objective necessitates worsening other objectives. Then, we propose a novel decomposition-based policy learning (DPPL) method, which involves (1) introducing a set of preference vectors, (2) dividing the whole optimization problem into several subproblems based on the preference vectors, and (3) ultimately achieving different Pareto solutions for the objectives by solving these subproblems. Compared with the linear weighting method, it can obtain Pareto optimal solutions in non-convex regions and is applicable to cases where multiple objectives are interrelated.

While the proposed DPPL method can find Pareto optimal policies, it necessitates specifying a set of preference vectors in advance. In practical applications, decision-makers may encounter the challenge of determining which preference vector to choose. To mitigate this concern, we further theoretically transform the optimization problem in DPPL into an $\varepsilon$-constraint problem. This transformation can assist decision-makers in better understanding and selecting preference vectors.

The contributions of this paper are summarized as follows.

• We formulate the policy learning problem of balancing multiple long-term and short-term rewards as a multi-objective optimization problem and propose a decomposition-based Pareto policy learning (DPPL) method to obtain a set of Pareto optimal policies.

• We theoretically establish the connection between the DPPL method and the $\epsilon$-constraint problem, offering an intuitive interpretation of preference vectors and guiding their selection.

• We conduct extensive experiments to demonstrate the effectiveness of the proposed method.

## 2   Problem Formulation

Throughout, we employ bold letters for vectors, uppercase letters for random variables, and lowercase letters for their realization values.

### 2.1   Notation

We introduce notations to delineate short-term and long-term causal effects. Let $A$ denote the binary treatment indicator, where $A = 1$ represents the treated group and $A = 0$ represents the control group. $\boldsymbol{X}$ represents the features observed, $\boldsymbol{S} = (S_1, ..., S_I) \in \mathbb{R}^I$ and $\boldsymbol{Y} = (Y_1, ..., Y_J) \in \mathbb{R}^J$ represent the vector of short-term and long-term outcomes, respectively. Both short-term and long-term outcomes are observed after the treatment $A$, and associations among them may exist.

Utilizing the potential outcome framework [11], we denote $\boldsymbol{S}(a) = (S_1(a), ..., S_I(a))$ and $\boldsymbol{Y}(a) = (Y_1(a), ..., Y_J(a))$ for $a = 0, 1$ as the potential short-term and long-term outcomes under treatment $A = a$, respectively. We assume that larger short-term and long-term outcomes are preferable. The observed short-term and long-term outcomes $\boldsymbol{S}$ and $\boldsymbol{Y}$ correspond to the potential outcomes of the actual treatment, that is, $\boldsymbol{S} = \boldsymbol{S}(A)$ and $\boldsymbol{Y} = \boldsymbol{Y}(A)$.

In real-world applications, long-term outcomes often suffer from missing due to prolonged follow-up periods and budget constraints. In contrast, collecting short-term outcomes is more manageable. Therefore, we presume that all short-term outcomes $\boldsymbol{S}$ are observable, while long-term outcomes $\boldsymbol{Y}$ may be subject to missing. Let $\boldsymbol{R} = (R_1, ..., R_J) \in \{0, 1\}^J$ denote the indicator for observing the long-term outcome $\boldsymbol{Y}$, where $R_j = 1$ indicates that $Y_j$ is observed and $R_j = 0$ indicates that $Y_j$ is missing. The missingness of $\boldsymbol{Y}$ would lead to identifiability and estimation problems [12–23].

### 2.2   Formulation

In this article, we aim to learn the Pareto optimal policy for balancing multiple correlated short-term and long-term rewards, which has a wide range of application scenarios [1, 6, 8, 24]. Let $\pi : \mathcal{X} \rightarrow \{0, 1\}$ be a policy that maps from the individual context $\boldsymbol{X} = \boldsymbol{x}$ to the treatment space

$\{0, 1\}$. For a given policy $\pi(\boldsymbol{\theta}) = \pi(\boldsymbol{X}, \boldsymbol{\theta})$ parameterized by $\boldsymbol{\theta}$, the policy values for the $i$-th short-term outcome $S_i$ and the $j$-th long-term outcome $Y_j$ are defined as,

$$\mathcal{V}(\boldsymbol{\theta}; s_i) = \mathbb{E}[\pi(\boldsymbol{\theta})S_i(1) + (1 - \pi(\boldsymbol{\theta}))S_i(0)], \quad i = 1, ..., I$$
$$\mathcal{V}(\boldsymbol{\theta}; y_j) = \mathbb{E}[\pi(\boldsymbol{\theta})Y_j(1) + (1 - \pi(\boldsymbol{\theta}))Y_j(0)], \quad j = 1, ..., J,$$

which are the $i$-th short-term reward and the $j$-the long-term reward induced by the policy $\pi(\boldsymbol{\theta})$.

Conventionally, we convert maximization problems to minimization problems. Let $\bar{\mathcal{V}}(\boldsymbol{\theta}; s_i) = -\mathcal{V}(\boldsymbol{\theta}; s_i), \bar{\mathcal{V}}(\boldsymbol{\theta}; y_j) = -\mathcal{V}(\boldsymbol{\theta}; y_j)$. The trade-off among multiple correlated long-term and short-term rewards can be formulated as a multi-objective optimization (MOP) problem given by

$$\min_{\boldsymbol{\theta}} \bar{\boldsymbol{\mathcal{V}}}(\boldsymbol{\theta}) = (\bar{\mathcal{V}}(\boldsymbol{\theta}; s_1), \cdots, \bar{\mathcal{V}}(\boldsymbol{\theta}; s_I), \bar{\mathcal{V}}(\boldsymbol{\theta}; y_1), \cdots, \bar{\mathcal{V}}(\boldsymbol{\theta}; y_J))$$
$$\triangleq (\bar{\mathcal{V}}_1(\boldsymbol{\theta}), \bar{\mathcal{V}}_2(\boldsymbol{\theta}), \cdots, \bar{\mathcal{V}}_M(\boldsymbol{\theta})) \tag{1}$$

where $M = I + J$ and the symbol $\triangleq$ means 'denoted as'. Generally, there is no single solution that can simultaneously optimize all objectives in problem (1) and thus we resort to the Pareto optimality. This concept is employed to define the optimal solutions for the MOP problem.

**Definition 1.** *(Pareto optimality)*

*(a) Pareto dominance. For two points $\boldsymbol{\theta}^1, \boldsymbol{\theta}^2$. $\boldsymbol{\theta}^1$ dominates $\boldsymbol{\theta}^2$ if and only if $\bar{\mathcal{V}}_m(\boldsymbol{\theta}^1) \leq \bar{\mathcal{V}}_m(\boldsymbol{\theta}^2), \forall m \in \{1, ..., M\}$ and $\bar{\mathcal{V}}_{m'}(\boldsymbol{\theta}^1) < \bar{\mathcal{V}}_{m'}(\boldsymbol{\theta}^2), \exists m' \in \{1, ..., M\}$*

*(b) Pareto optimality. $\boldsymbol{\theta}^*$ is a Pareto optimal point if there is no other solution $\hat{\boldsymbol{\theta}}$ that dominates $\boldsymbol{\theta}^*$.*

Pareto optimality refers to a condition where improving one objective comes at the expense of worsening other objectives. The collection of Pareto optimal solutions is called the Pareto set. Our goal is to derive the set of Pareto optimal solutions (or Pareto optimal policies), each of them providing a distinct optimal trade-off among all objectives.

## 2.3 Identification and Estimation of Short-term and Long-term Rewards

The long-term and short-term rewards are causal parameters that cannot be identified without imposing causal assumptions [25–27]. Therefore, before seeking the Pareto optimal solutions for balancing multiple long-term and short-term rewards, it is necessary to consider the identification and estimation of long-term and short-term rewards. The proposed method is based on Assumptions 1 and 2 below.

**Assumption 1** (Strong Ignorability)**.**

*(a) $(\boldsymbol{S}(a), \boldsymbol{Y}(a)) \perp\!\!\!\perp A \mid \boldsymbol{X}$ for $a = 0, 1$;*

*(b) $0 < e(\boldsymbol{x}) \triangleq \mathbb{P}(A = 1 \mid \boldsymbol{X} = \boldsymbol{x}) < 1$ for all $\boldsymbol{x}$.*

Assumption 1(a) suggests that, given the feature $\boldsymbol{X}$, treatment assignment $A$ is independent of the potential outcomes $\boldsymbol{S}(a)$ and $\boldsymbol{Y}(a)$. This implies that confounding bias between the treatment $A$ and the short/long-term outcomes $(\boldsymbol{S}(a), \boldsymbol{Y}(a))$ can be eliminated by conditioning on $\boldsymbol{X}$ [28]. Assumption 1(b) ensures that for the subpopulation of $\boldsymbol{X} = \boldsymbol{x}$, units with both $A = 1$ and $A = 0$ exist. These assumptions are widely used in causal inference [11, 27, 29–35].

In addition to confounding bias, we also need to address the selection bias induced by the missingness of long-term outcomes [8]. Thus, we further invoke the Assumption 2.

**Assumption 2** (Missing Mechanism of Long-term Outcome)**.** *For $a = 0, 1$ and $j = 1, ..., J$,*

*(a) $R_j \perp\!\!\!\perp Y_j(a) \mid \boldsymbol{X}, \boldsymbol{S}(a), A = a$;*

*(b) $0 < r_j(\boldsymbol{x}, a, \boldsymbol{s}) \triangleq \mathbb{P}(R_j = 1 \mid \boldsymbol{X} = \boldsymbol{x}, A = a, \boldsymbol{S} = \boldsymbol{s})$.*

Assumption 2(a) can be reformulated as $R_j \perp\!\!\!\perp Y_j \mid (\boldsymbol{X}, \boldsymbol{S}, A)$, which means that $R_j$ relies only on the observed variables $(\boldsymbol{X}, A, \boldsymbol{S})$. This assumption also ensures that $\mathbb{P}(Y_j = y | \boldsymbol{X}, \boldsymbol{S}, A, R_j = 1) = \mathbb{P}(Y_j = y | \boldsymbol{X}, \boldsymbol{S}, A, R_j = 0)$. This implies that we can utilize the available data to draw conclusions about the missing long-term outcome. Assumption 2(b) assumes that the long-term outcome for each unit has a non-zero probability of being observed. Assumptions 1 and 2 ensures the identifiability of $\mathcal{V}(\boldsymbol{\theta}; s_i)$ and $\mathcal{V}(\boldsymbol{\theta}; y_j)$, as shown in Lemma 1.

**Lemma 1** (Identifiability of Short-term and Long-term Rewards). *For $i = 1, ..., I$ and $j = 1, ..., J$,*

*(a) under Assumptions 1, the $i$-th short-term reward $\mathcal{V}(\boldsymbol{\theta}; s_i)$ is identifiable.*

*(b) under Assumptions 1-2, the $j$-th long-term reward $\mathcal{V}(\boldsymbol{\theta}; y_j)$ is identifiable.*

When we have access to only one short-term outcome and one long-term outcome, Lemma 1 reduces to the identifiability result presented in [8]. In this article, our focus is on achieving the Pareto optimal policy for multiple short-term and long-term rewards. Therefore, for the estimation of $\mathcal{V}(\boldsymbol{\theta}; s_i)$ and $\mathcal{V}(\boldsymbol{\theta}; y_j)$, we defer it to Appendix A.

## 3 Pareto Policy Learning for Balancing Short-Term and Long-Term Rewards

In this section, we aim to learn Pareto optimal policies for the MOP problem (1). Section 3.1 gives the motivation for this work and Section 3.2 introduces the proposed policy learning approach. In Section 3.3, we theoretically establishe the connection between the linear weighting method, the MOP problem for a given preference vector, and the $\varepsilon$-constraint problem. This connection offers an intuitive interpretation and guides practitioners in selecting the preference vector.

### 3.1 Motivation

For seeking the optimal policy for balancing short-term and long-term rewards, previous work [8] adopted the linear weighting method. Specifically, the authors formulate the goal as

$$\min_{\boldsymbol{\theta}} \bar{\mathcal{V}}(\boldsymbol{\theta}) = \sum_{m=1}^{M} \omega_m \bar{\mathcal{V}}_m(\boldsymbol{\theta}), \tag{2}$$

where $\omega_m$ is the pre-specified weight for the $m$-th objective. The objective function in the optimization problem (2) is merely a linear combination of multiple objectives from the MOP problem (1). Due to its intuitiveness and simplicity, the traditional linear weighting method is commonly used for solving MOP or multi-task learning problems [36–38].

The linear weighting method simply combines multiple objectives into a single surrogate objective through weighted summation. While simple, it has several limitations. First, the optimal solution is found only in convex regions and not in non-convex regions [9]. Second, an optimal solution can only be achieved if the objectives are independent of each other. That is, if some objectives are interrelated, only a suboptimal solution can be obtained [10]. Thus, it does not guarantee the superiority of the solution or its solution may deviate from the Pareto optimal solution.

To overcome the limitations of the linear weighting method in [8], we first introduce a decomposition-based multi-objective optimization algorithm to achieve the Pareto optimal policy. However, this algorithm relies on pre-specified preference vectors, which are used to express a decision maker's degree of preference for multiple conflicting objectives. In practice, the explanation and selection of preference vectors is a challenging problem. To further tackle this issue, we establish a theoretical relationship between preference vectors and the $\varepsilon$-constraint method [39]. This relationship provides a clear interpretation on preference vectors, assisting in selecting more suitable ones.

### 3.2 Pareto Policy Learning for the MOP Problem

We introduce the decomposition-based Pareto policy learning (DPPL) method, which can generate the Pareto set containing policies that are optimum from a trade-off perspective. The main idea of the DPPL method is to first decompose the original MOP problem into several constrained subproblems based on a predefined set of preference vectors, and then obtain a set of Pareto optimal policies by solving these subproblems in parallel [40].

For obtaining the Pareto optimal policy for balancing $M$ short-term and long-term objectives, first, we are given a set of $K$ preference vectors $\{\boldsymbol{u}_1, \boldsymbol{u}_2, ..., \boldsymbol{u}_K\}$ in $\mathbb{R}_+^M$. Each element of a preference vector specifies the importance of the corresponding short-term or long-term reward. For each preference vector $\boldsymbol{u}_k$, the corresponding subproblem is given as

$$\min_{\boldsymbol{\theta}} \bar{\boldsymbol{\mathcal{V}}}(\boldsymbol{\theta}) = (\bar{\mathcal{V}}_1(\boldsymbol{\theta}), \bar{\mathcal{V}}_2(\boldsymbol{\theta}), \cdots, \bar{\mathcal{V}}_M(\boldsymbol{\theta}))$$

$$s.t. \ \mathcal{G}_{k'}(\boldsymbol{\theta}) = (\boldsymbol{u}_{k'} - \boldsymbol{u}_k)^T \bar{\boldsymbol{\mathcal{V}}}(\boldsymbol{\theta}) \leq 0, \ \forall \, k' = 1, ..., K, \tag{3}$$

where $\mathcal{G}_{k'}(\boldsymbol{\theta}_t) \leq 0$ means that objective space[2] of the subproblem is restricted in the subregion $\Omega_k$, which is defined by $\Omega_k = \{\boldsymbol{v} \in R_+^M | \boldsymbol{u}_{k'}^T \boldsymbol{v} \leq \boldsymbol{u}_k^T \boldsymbol{v}, \forall\, k' = 1, ..., K\}$. Geometrically speaking, $\Omega_k$ represents the set of $\boldsymbol{v}$ that forms the smallest acute angle with $\boldsymbol{u}_k$, which means that the optimal solution of the subproblem can be obtained by only searching the subregion. The preference vectors divide the objective space into different subregions.

Solving the subproblem (3) involves the following two steps:

- **Step (a)**. Find a reasonable initial solution $\boldsymbol{\theta}_0$. Specifically, we first randomly generate a solution $\boldsymbol{\theta}_r$ in the full decision space[3], and then iteratively update it with the rule $\boldsymbol{\theta}_{r_{t+1}} = \boldsymbol{\theta}_{r_t} + \eta_r \boldsymbol{d}_{r_t}$, where $\eta_r$ is the step size. For a given $\boldsymbol{\theta}_{r_t}$, the descent direction $\boldsymbol{d}_{r_t}$ is updated by solving (4).

$$(\boldsymbol{d}_{r_t}, \alpha_{r_t}) = \arg \min_{\boldsymbol{d} \in \mathbb{R}^n, \alpha \in \mathbb{R}} \alpha + \frac{1}{2}||\boldsymbol{d}||^2, \ s.t. \ \nabla \mathcal{G}_{k'}(\boldsymbol{\theta}_{r_t})^T \boldsymbol{d} \leq \alpha, \ k' \in \mathcal{I}(\boldsymbol{\theta}_{r_t}). \tag{4}$$

where $\mathcal{I}(\boldsymbol{\theta}_{r_t}) = \{k' | \mathcal{G}_{k'}(\boldsymbol{\theta}_{r_t}) \geq 0, k' = 1, ..., K\}$ is index set of all activated constraints, which means $\bar{\mathcal{V}}(\boldsymbol{\theta}_{r_t})$ not in $\Omega_k$. The problem (4) aims to find the descent direction $\boldsymbol{d}_{r_t}$ for each iteration $t$ and then obtain the initial solution $\boldsymbol{\theta}_0$ such that $\bar{\mathcal{V}}(\boldsymbol{\theta}_0)$ in $\Omega_k$.

- **Step (b)**. Solving the subproblem (3). The descent direction $\boldsymbol{d}_t$ for the $t$-th iteration is obtained by

$$(\boldsymbol{d}_t, \alpha_t) = \arg \min_{d \in \mathbb{R}^n, \alpha \in \mathbb{R}} \alpha + \frac{1}{2}||\boldsymbol{d}||^2$$
$$s.t. \quad \nabla \bar{\mathcal{V}}_m(\boldsymbol{\theta}_t)^T \boldsymbol{d} \leq \alpha, m = 1, ..., M.$$
$$\nabla \mathcal{G}_{k'}(\boldsymbol{\theta}_t)^T \boldsymbol{d} \leq \alpha, k' \in \mathcal{I}_\epsilon(\boldsymbol{\theta}_t), \tag{5}$$

where $\mathcal{I}_\epsilon(\boldsymbol{\theta}) = \{k' | \mathcal{G}_{k'}(\boldsymbol{\theta}) \geq -\epsilon\}$, and the threshold $\epsilon$ is a slack variable used to deal with the solutions near the constraint boundary. We further transform it into a dual problem which will greatly reduce the dimension of decision space. Based on the KKT conditions, we have $\boldsymbol{d}_t = -(\sum_{m=1}^M \lambda_m \nabla \bar{\mathcal{V}}_m(\boldsymbol{\theta}_t) + \sum_{k' \in \mathcal{I}_\epsilon(\boldsymbol{\theta})} \beta_{k'} \nabla \mathcal{G}_{k'}(\boldsymbol{\theta}_t))$. Therefore, the dual problem is given as

$$\max_{\lambda_m, \beta_{k'}} -\frac{1}{2}|| \sum_{m=1}^M \lambda_m \nabla \bar{\mathcal{V}}_m(\boldsymbol{\theta}_t) + \sum_{k' \in I_\epsilon(\boldsymbol{\theta})} \beta_{k'} \nabla \mathcal{G}_{k'}(\boldsymbol{\theta}_t)||^2$$
$$\tag{6}$$
$$s.t. \ \sum_{m=1}^M \lambda_m + \sum_{k' \in I_\epsilon(\boldsymbol{\theta})} \beta_{k'} = 1, \lambda_m \geq 0, \beta_{k'} \geq 0, \forall m = 1, ..., M, \forall\, k' \in \mathcal{I}_\epsilon(\boldsymbol{\theta}).$$

where $\lambda_m \geq 0$ and $\beta_{k'} \geq 0$ are the Lagrange multipliers for the linear inequality constraints.

Step (a) is to find an initial solution $\boldsymbol{\theta}_0$ that is restricted in a subregion of the subproblem (3), and once a feasible solution is found or a predetermined number of iterations is reached, the step stops. For an given initial solution $\boldsymbol{\theta}_0$, Step (b) is to find the optimal solution $\boldsymbol{\theta}^*$ for the subproblem (3). We summarize the proposed policy learning approach in Appendix B.

**Lemma 2** ([41]). *Let $(\boldsymbol{d}_t, \alpha_t)$ be the solution to the $t$-th iteration of problem* (5).

*(a) If $\boldsymbol{\theta}_t$ is Pareto optimal restricted on $\Omega_k$, then $\boldsymbol{d}_t = 0 \in \mathbb{R}^m$ and $\alpha_t = 0$.*

*(b) If $\boldsymbol{\theta}_t$ is not Pareto optimal restricted on $\Omega_k$. then*

$$\alpha_t \leq -(1/2)||d_t||^2 < 0,$$
$$\nabla \bar{\mathcal{V}}_m(\boldsymbol{\theta}_t)^T \boldsymbol{d}_t \leq \alpha_t, m = 1, ..., M \tag{7}$$
$$\nabla \mathcal{G}_{k'}(\boldsymbol{\theta}_t)^T \boldsymbol{d}_t \leq \alpha_t, k' \in \mathcal{I}_\epsilon(\boldsymbol{\theta}_t).$$

Lemma 2(a) implies that at the $t$-th iteration, no direction ($\boldsymbol{d}_t = 0$) can simultaneously improve the performance for all objectives, confirming that the solution $\boldsymbol{\theta}_t$ satisfies Pareto optimality. Lemma 2(b) suggests that if $\boldsymbol{\theta}_t$ does not meet Pareto optimality, then the descent direction $\boldsymbol{d}_t \neq 0$ serves as the descent direction for all objectives, such that the solution of the next iteration is closer to the Pareto optimal solution. Thus, Lemma 2 demonstrates that we always attain Pareto optimal solutions for each subproblem using the update rule $\boldsymbol{\theta}_{t+1} = \boldsymbol{\theta}_t + \eta_r \boldsymbol{d}_t$. By solving all subproblems, we can acquire a diverse set of Pareto optimal solutions (or policies) confined to different subregions, even when the multiple objectives are correlated.

---

[2] $\bar{\mathcal{V}}(\boldsymbol{\theta})$ is the objective vector, and the space spanned by the objective vectors is called the objective space $\Omega$.
[3] The parameter vector $\boldsymbol{\theta}$ represents the decision variable and the space spanned it is called the decision space.

### 3.3 Deep Analysis of the Preference Vector

The DPPL method in Section 3.2 requires a set of pre-specified preference vectors, posing challenges in practical applications where selecting suitable preference vectors is non-trivial. To mitigate this problem, we provide a practical method for decision-makers to select appropriate preference vectors by theoretically establishing the connection between the DPPL method and the $\varepsilon$-constraint problem.

We first give a brief introduction to the $\varepsilon$-constraint problem [10], which is defined as follows,

$$\min_{\boldsymbol{\theta}} \bar{\mathcal{V}}_l(\boldsymbol{\theta}), \ \text{ s. t. } \bar{\mathcal{V}}_m(\boldsymbol{\theta}) \leq \varepsilon_m \text{ for all } m = 1, \ldots, M, m \neq l, \tag{8}$$

where $\varepsilon_m$ is pre-specified threshold. Compared to the MOP problem (1) and the linear weighting objective (2), a notable advantage of the $\varepsilon$-constraint problem is its interpretation on the threshold $\varepsilon_m$, which represents the maximum acceptable value (i.e., the acceptable worst-case scenario) for the $m$-th objective. In contrast, the weights and the preference vectors in problems (1) and (2) are not straightforward for relating the resulting values of objectives. Thus, if we can establish the connection between $\boldsymbol{\varepsilon} = (\varepsilon_1, ..., \varepsilon_M)$ and the preference vector $\boldsymbol{u}_k$, then we can provide powerful guidance for choosing appropriate preference vectors.

**Theorem 1.** *For the preference vector* $\boldsymbol{u}_k = (u_{k1}, ..., u_{kM})$ *in problem* (1)*, the weights* $\boldsymbol{\omega} = (\omega_1, ...., \omega_M)$ *in problem* (2)*, and the thresholds* $\boldsymbol{\varepsilon}$ *in problem* (8)*, the following statements hold:*

*(a) the connection between* $\boldsymbol{\varepsilon}$ *and* $\boldsymbol{\omega}$ *is given as*

$$\varepsilon_m = -\mathbb{E}[\mathbb{I}(\tau_l(\boldsymbol{X}) + \frac{\omega_m}{\omega_l}\tau_m(\boldsymbol{X}) > 0) \cdot \tau_m(\boldsymbol{X}) + h_m(\boldsymbol{X})], \text{ for } m = 1 \cdots M, \text{and } m \neq l, \tag{9}$$

*where* $\tau_m(\boldsymbol{X})$ *is the conditional average causal effects for* $m$-*th short/long-term outcome,*

$$\tau_m(\boldsymbol{X}) = \begin{cases} \mathbb{E}[S_i(1) - S_i(0)|\boldsymbol{X}], & \text{if } \omega_m \text{ is the weight of } \bar{\mathcal{V}}(\boldsymbol{\theta}, s_i), \\ \mathbb{E}[Y_j(1) - Y_j(0)|\boldsymbol{X}], & \text{if } \omega_m \text{ is the weight of } \bar{\mathcal{V}}(\boldsymbol{\theta}, y_j), \end{cases}$$

$\mathbb{I}(\cdot)$ *is the indicator function, and*

$$h_m(\boldsymbol{X}) = \begin{cases} \mathbb{E}[S_i(0)|\boldsymbol{X}], & \text{if } \omega_m \text{ is the weight of } \bar{\mathcal{V}}(\boldsymbol{\theta}, s_i), \\ \mathbb{E}[Y_j(0)|\boldsymbol{X}, \boldsymbol{S}, R_j = 1], & \text{if } \omega_m \text{ is the weight of } \bar{\mathcal{V}}(\boldsymbol{\theta}, y_j). \end{cases}$$

*(b) the connection between* $\boldsymbol{\omega}$ *and* $\boldsymbol{u}_k$ *is given as*

$$\omega_m = \lambda_m + \sum_{k' \in \mathcal{I}_\epsilon(\boldsymbol{\theta})} \beta_{k'}(\boldsymbol{u}_{k'm} - \boldsymbol{u}_{km}), \text{ for } m = 1, \cdots, M, \tag{10}$$

*where* $\lambda_m$ *and* $\beta_{k'}$ *are defined in Eq.* (6)*, and* $\mathcal{I}_\epsilon(\boldsymbol{\theta}) = \{k'|\mathcal{G}_{k'}(\boldsymbol{\theta}) \geq -\epsilon\}$ *defined in Eq.* (4).

Theorem 1 (see Appendix C for proofs) establishes a link between the preference vector $\boldsymbol{u}_k$ and $\boldsymbol{\varepsilon}$ through $\boldsymbol{\omega}$ in scenarios involving multiple long-term and short-term objectives. Specifically, Theorem 1(a) shows how to estimate the threshold $\boldsymbol{\varepsilon}$ for given weights $\boldsymbol{\omega}$, and Theorem 1(b) shows how to assign weights $\boldsymbol{\omega}$ via preference vectors $\boldsymbol{u}_k$. This means that for the subproblem determined by preference vectors $\boldsymbol{u}_k$, we can ascertain the maximum acceptable threshold $\boldsymbol{\varepsilon}$ based on Theorem 1, thereby offering an intuitive interpretation of the preference vector $\boldsymbol{u}_k$.

There are several practical implications with Theorem 1. On one hand, it assists decision-makers in better understanding and selecting preference vectors in practical applications. In practice, we can initially pre-specify a set of preference vectors $\{\boldsymbol{u}_1, \boldsymbol{u}_2, ..., \boldsymbol{u}_K\}$ in $\mathbb{R}_+^M$, then derive the weights $\boldsymbol{\omega}$ corresponding to each preference vector $\boldsymbol{u}_k$ through Eq. (10), and finally substitute the obtained weight $\boldsymbol{\omega}$ into Eq. (9) to calculate the threshold $\boldsymbol{\varepsilon}$. Leveraging the intuitive interpretability of the threshold $\boldsymbol{\varepsilon}$, decision-makers can select the appropriate preference vectors according to their specific requirements. On the other hand, it also provides guidance for specifying $\boldsymbol{\varepsilon}$ in the $\varepsilon$-constraint problem (8). Inappropriate selection of $\boldsymbol{\varepsilon}$ for this problem may result in an empty feasible region, yielding empty solutions. By utilizing a set of preference vectors, we can efficiently screen out some reasonable choices of $\boldsymbol{\varepsilon}$ and reduce the cumbersome trial-and-error process of testing different $\boldsymbol{\varepsilon}$.

In conclusion, by establishing the connection between the DPPL method and the $\varepsilon$-constraint problem, we can harness the advantages of both methods while mitigating their respective weaknesses.

# 4 Experiments

**Datasets.** Following the previous studies [8], we use two widely used datasets: IHDP and JOBS, for evaluating the performance of the proposed method. The IHDP dataset explores the effectiveness of high-quality home visiting in promoting children's future cognitive development and covers a sample of 747 units, including 139 treated and 608 controlled. In addition, the dataset has 25 characteristics that provide a comprehensive picture of the children and their mothers. The second dataset, JOBS, explores the effects of job training on income and employment status. It consists of 2,570 units (237 treated, 2,333 controlled), with 17 covariates. Note that each unit in both datasets has only one observed outcome from a single treatment, and neither dataset collects long-term outcomes.

**Simulating Outcome.** Consider the case of one long-term reward and one short-term reward. Following the previous data-generation mechanisms [1, 42], for the $n$-th unit ($n = 1, ..., N$), we simulate the potential short-term outcomes $S(0)$ and $S(1)$ as follows:

$$S_n(0) \sim \text{Bern}(\sigma(w_0 X_n + \epsilon_{0,n})), \quad S_n(1) \sim \text{Bern}(\sigma(w_1 X_n + \epsilon_{1,n})),$$

where $\sigma(\cdot)$ is the sigmoid function, $w_0 \sim \mathcal{N}_{[-1,1]}(0, 1)$ follows a truncated normal distribution, $w_1 \sim \text{Unif}(-1, 1)$ follows a uniform distribution, $\epsilon_{0,n} \sim \mathcal{N}(\mu_0, \sigma_0)$ and $\epsilon_{1,n} \sim \mathcal{N}(\mu_1, \sigma_1)$. We set $\mu_0 = 1, \mu_1 = 3$ and $\sigma_0 = \sigma_1 = 1$ for the IHDP dataset, and we set $\mu_0 = 0, \mu_1 = 2$ and $\sigma_0 = \sigma_1 = 1$ for the JOBS dataset. For generating long-term potential outcomes $Y(0)$ and $Y(1)$, we introduce the time step $t$: we set the initial value at time step 0 as: $Y_{0,n}(0) = S_n(0), Y_{0,n}(1) = S_n(1)$, then generate $Y_{t,n}(0), Y_{t,n}(1)$ according to the following equation and we eventually regard the outcome at the last time step $T$ as the long-term outcome, $Y_n(0) = Y_{T,n}(0), Y_n(1) = Y_{T,n}(1)$.

$$Y_{t,n}(0) \sim \text{Bern}(\sigma(\beta_0 X_n) + C \sum_{t'=0}^{t-1} Y_{t',n}(0)) + \epsilon_{0,n}, \; Y_{t,n}(1) \sim \text{Bern}(\sigma(\beta_1 X_n) + C \sum_{t'=0}^{t-1} Y_{t',n}(0)) + \epsilon_{1,n},$$

where $\beta_0$ is randomly sampled from $\{0, 1, 2, 3, 4\}$ with probabilities $\{0.5, 0.2, 0.15, 0.1, 0.05\}$, $\beta_1 \sim 4 \cdot \mathcal{N}_{[0,4]}(0, 1)$, and $C = 1/T$ is a scaling factor. For $\epsilon_{0,n}$ and $\epsilon_{1,n}$, we set $\mu_0 = \mu_1 = 0, \sigma_0 = 1$ and $\sigma_1 = 3$ for the IHDP dataset and set $\mu_0 = \mu_1 = 0, \sigma_0 = 1$ and $\sigma_1 = 1$ for the JOBS dataset.

Assumption 2 shows that observing indicator $R$ depends on the feature $\boldsymbol{X}$, the treatment $A$, and short-term outcome $S$. For a given missing rate $r$, we select the missing indexes for $Y$ and derive the missing indicator $R$ according to the following criterion: calculate the $m_n = 1/D \sum_{d=1}^{D}(X_{nd} + s_n), n = 1, \cdots, N$, and choose the index of the row with the smallest $rN$ values in $\{m_n, n = 1, \cdots, N\}$ as the missing indexes. $D$ is the feature dimension and $N$ is the sample size.

**Experimental Details.** In this paper, preference vectors are used to quantify an individual's preference for different objectives in the multi-objective optimization problem. For the case of two-objective, we randomly generate 10 unit preference vectors $(\boldsymbol{u}_1, \boldsymbol{u}_2, \cdots, \boldsymbol{u}_{10})$, where $\boldsymbol{u}_k = (u_{k1}, u_{k2})$, $u_{k1} = cos(t_k), u_{k2} = sin(t_k), t_k \in (0, 1)$, which implies that the $L_2$-norm of the preference vectors is 1, ensuring the consistency and comparability of the preference measures. $u_{k1}$ and $u_{k2}$ are the preferences for the short-term objective and the long-term objective, respectively. Each component of the preference vector $\boldsymbol{u}_k$ represents the strength or importance of the decision maker's preference for different objectives. Preference vectors are used as weights in the linear weighting method, whereas our method uses them to divide the original problem (1) into several subproblems.

**Evaluation Metrics.** We measure the performance of our proposed method by three metrics: long and short-term rewards, the variance of long and short-term rewards, and the change in welfare. Formally, the short-term reward of the learned policy $\hat{\pi}(\boldsymbol{X}, \boldsymbol{\theta})$ is $\hat{\mathcal{V}}(\boldsymbol{\theta}; s) = \sum_{n=1}^{N}[\hat{\pi}(X_n, \boldsymbol{\theta})S_n(1) + (1 - \hat{\pi}(X_n, \boldsymbol{\theta}))S_n(0)]$, the long-term reward is $\hat{\mathcal{V}}(\boldsymbol{\theta}; y) = \sum_{n=1}^{N}[\hat{\pi}(X_n, \boldsymbol{\theta})Y_n(1) + (1 - \hat{\pi}(X_n, \boldsymbol{\theta}))Y_n(0)]$. Similar as [42, 43], the welfare changes are defined as $\Delta W_s = \sum_{n=1}^{N}[(S_n(1) - S_n(0)) \cdot \hat{\pi}(X_n, \boldsymbol{\theta})]$ for the short-term reward, $\Delta W_y = \sum_{n=1}^{N}[(Y_n(1) - Y_n(0)) \cdot \hat{\pi}(X_n, \boldsymbol{\theta})]$ for the long-term reward, $\Delta W = 0.5\Delta W_s + 0.5\Delta W_y$ for the overall balanced-base reward. Among these metrics, $\Delta W$ is the most critical here, as it directly measures the balance reward achieved by the learned policy.

**Policy learning with short-term and short-term reward.** We choose MLP as the policy model $\pi(\boldsymbol{\theta})$, and we average over 50 independent trials of policy learning with the short-term and long-term reward in IHDP and JOBS. We fix the missing ratio $r = 0.2$ and the time step $T = 4$. We measure the uncertainty of the model by calculating the variance of the long and short-term reward over 50 experiments, and a smaller variance means a more stable model performance.

**Performance Comparison.** From our previous analyses, the linear weighting method generally achieves the sub-optimal policies. The proposed DPPL method can generate a set of Pareto optimal policies. First, for the long-term reward, the short-term reward, and $\Delta W$, it is not surprising to observe that for most of the preference vectors, DPPL's solutions have better performance. Second, for the variance, our method performs more stable in 50 experiments. Because we will divide the original problem into several subproblems according to preference vectors, and then solve the subproblems in a relatively small subregion to obtain the Pareto optimal solution, whereas the linear weighting method searches the entire objective space. The associated results are displayed in Table 1. More experimental results with missing ratio $r = 0.3$ are given in Appendix D.

Table 1: Comparison of our method (OURS) and linear weighting method (LW) on IHDP and JOBS, with Short-Term Reward (S-REWARDS) and Long-Term Reward (L-REWARDS), $\Delta W$ and Variance (S-VAR and L-VAR) as evaluation metrics. The best result is bolded.

| IHDP | S-REWARDS | | L-REWARDS | | $\Delta W$ | | S-VAR | | L-VAR | |
|---|---|---|---|---|---|---|---|---|---|---|
| PREFERENCE VECTOR | OURS | LW | OURS | LW | OURS | LW | OURS | LW | OURS | LW |
| 1 (1.00, 0.00) | **522.840** | 520.860 | **386.221** | 383.950 | **39.432** | 37.307 | 14.573 | **12.841** | **52.326** | 56.093 |
| 2 (0.98, 0.17) | **521.820** | 524.660 | 382.774 | **387.102** | 37.199 | **40.782** | 13.275 | **11.079** | **54.181** | 59.895 |
| 3 (0.94, 0.34) | **523.000** | 521.840 | 372.418 | **394.386** | 32.610 | **43.014** | **11.588** | 13.578 | **50.138** | 62.584 |
| 4 (0.86, 0.50) | **521.060** | 519.680 | **382.419** | 379.174 | **36.641** | 34.328 | **11.512** | 13.299 | 52.165 | **48.457** |
| 5 (0.76, 0.64) | **523.620** | 519.840 | **391.296** | 390.413 | **42.360** | 40.028 | **12.729** | 15.594 | 55.883 | **48.308** |
| 6 (0.64, 0.76) | **521.460** | 517.420 | 387.015 | **390.206** | **39.139** | 38.714 | **14.217** | 15.479 | **46.342** | 56.219 |
| 7 (0.50, 0.87) | **523.800** | 514.480 | **383.424** | 381.321 | **38.514** | 32.802 | **12.797** | 18.758 | **55.118** | 55.167 |
| 8 (0.34, 0.94) | **521.360** | 516.800 | 373.307 | **400.510** | 32.235 | **43.556** | **11.701** | 18.618 | **50.002** | 60.847 |
| 9 (0.17, 0.98) | **522.240** | 515.040 | **397.640** | 396.214 | **42.842** | 40.529 | **12.913** | 18.543 | **57.288** | 59.071 |
| 10 (0.00, 1.00) | **523.600** | 516.780 | 387.933 | **390.387** | **40.668** | 38.485 | **11.531** | 21.079 | 60.714 | **54.246** |

| JOBS | S-REWARDS | | L-REWARDS | | $\Delta W$ | | S-VAR | | L-VAR | |
|---|---|---|---|---|---|---|---|---|---|---|
| PREFERENCE VECTOR | OURS | LW | OURS | LW | OURS | LW | OURS | LW | OURS | LW |
| 1 (1.00, 0.00) | **1613.140** | 1612.340 | **1230.918** | 1223.416 | **159.381** | 155.230 | **54.724** | 56.502 | **84.846** | 88.008 |
| 2 (0.98, 0.17) | **1618.400** | 1607.680 | 1219.517 | **1223.072** | **156.310** | 152.728 | **54.521** | 65.927 | **85.566** | 86.622 |
| 3 (0.94, 0.34) | **1614.800** | 1598.460 | 1220.316 | **1223.813** | **154.910** | 148.488 | **61.466** | 74.649 | **94.643** | 98.588 |
| 4 (0.86, 0.50) | **1612.880** | 1598.880 | 1217.305 | **1225.574** | **152.444** | 149.579 | **59.510** | 75.431 | 90.858 | **79.728** |
| 5 (0.77, 0.64) | **1613.160** | 1602.320 | **1233.604** | 1227.886 | **160.734** | 152.455 | **59.042** | 77.481 | **85.553** | 85.854 |
| 6 (0.64, 0.76) | **1612.380** | 1595.960 | 1218.100 | **1219.996** | **152.592** | 145.330 | **58.028** | 82.032 | 96.137 | **94.424** |
| 7 (0.50, 0.86) | **1608.860** | 1596.280 | 1224.763 | **1230.471** | **154.163** | 150.727 | **58.706** | 86.083 | **89.392** | 92.135 |
| 8 (0.34, 0.94) | **1613.600** | 1595.720 | **1232.958** | 1217.118 | **160.631** | 143.771 | **57.000** | 82.996 | **81.431** | 86.121 |
| 9 (0.17, 0.98) | **1614.840** | 1596.320 | **1225.607** | 1224.383 | **157.575** | 147.703 | **58.278** | 84.221 | 99.329 | **82.437** |
| 10 (0.00, 1.00) | **1610.380** | 1588.400 | **1228.679** | 1223.119 | **156.882** | 143.112 | **59.285** | 88.393 | 95.054 | **85.443** |

**Sensitivity Analysis.** We perform the sensitivity analysis of missing ratio $r$ and time step $T$ on JOBS. Our method achieves better performance in all missing rates $r = [0.2, 0.3, 0.4, 0.5]$ with $T = 4$, and $r = 0.2$ with time step $T = [4, 6, 8, 10]$. Our method stably outperforms the linear weighting method under varying $r$ and $T$, even in scenarios with a high missing ratio or a large time step. This further illustrates the effectiveness of our method. The associated results are displayed in Figure 1.

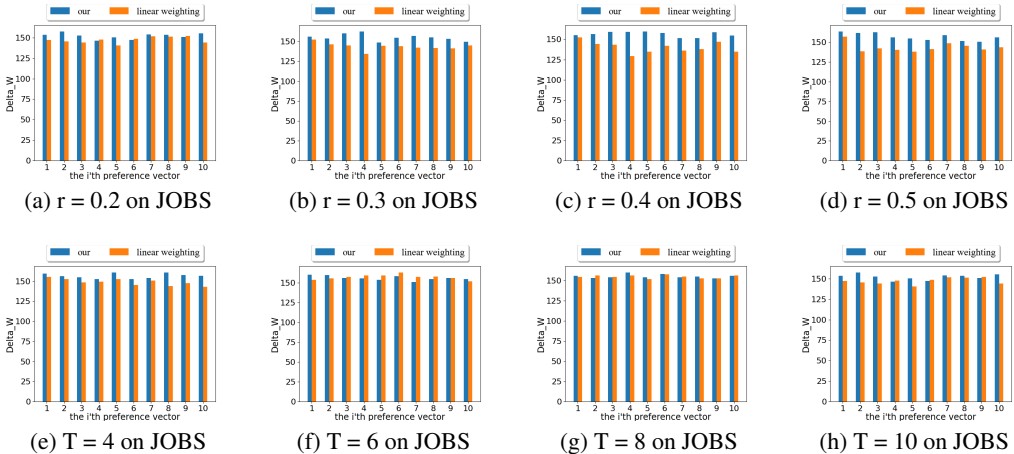

Figure 1: Comparison of two methods with different missing ratios $\{0.2, 0.3, 0.4, 0.5\}$ on JOBS

**Interpretation on preference vectors.** By Theorem 1, for the set of pre-specified preference vectors $(\boldsymbol{u}_1, \boldsymbol{u}_2, \cdots, \boldsymbol{u}_{10})$, we transform the optimization subproblem corresponding to each preference vector into the $\varepsilon$-constraint problem as $\min_{\boldsymbol{\theta}} \bar{\mathcal{V}}(\boldsymbol{\theta}; y), s.t. \bar{\mathcal{V}}(\boldsymbol{\theta}; s) \leq \varepsilon(< 0)$ or

$\max_{\boldsymbol{\theta}} \mathcal{V}(\boldsymbol{\theta}; y), s.t. \mathcal{V}(\boldsymbol{\theta}; s) \geq -\varepsilon$ and the threshold $-\varepsilon$ are shown in Table 2. This value of $-\varepsilon$ is the minimum value of the short-term reward that the decision maker can accept while maximizing the long-term reward. Our results show that as the second component of the preference vector increases, the value of $-\varepsilon$ shows a decreasing trend. In essence, this signifies that a decision-maker who emphasizes the long-term reward must necessarily loosen constraints on the short-term reward. In practice, decision makers can determine the threshold based on their specific needs for the short-term reward, and then select the most appropriate preference vector from the set of pre-specify preference vectors with the help of the intuitive interpretability of the threshold according to Table 2. More experimental results with different missing ratios $\{0.3, 0.4, 0.5\}$ are provided in Appendix D.

Table 2: The $\varepsilon$ values correspond to each preference vector in IHDP and JOBS datasets, where $T = 4$ and $r = 0.2$, obtained according to Theorem 1.

| Preference Vector | -$\varepsilon$ on IHDP | -$\varepsilon$ on JOBS | Preference Vector | -$\varepsilon$ on IHDP | -$\varepsilon$ on JOBS |
|---|---|---|---|---|---|
| (1.00, 0.00) | 0.820 | 0.878 | (0.00, 1.00) | 0.522 | 0.737 |
| (0.98, 0.17) | 0.827 | 0.875 | (0.17, 0.98) | 0.522 | 0.716 |
| (0.94, 0.34) | 0.826 | 0.868 | (0.34, 0.94) | 0.511 | 0.704 |
| (0.86, 0.50) | 0.833 | 0.868 | (0.50, 0.86) | 0.557 | 0.746 |
| (0.77, 0.64) | 0.741 | 0.865 | (0.64, 0.76) | 0.659 | 0.808 |

## 5   Related Work

**Estimation of long-term causal effects.** Assessing long-term causal effects is challenging due to the delayed long-term outcomes, posing significant difficulties in both identification and estimation. Recently, there has been increasing interest in using short-term surrogates to identify and estimate long-term causal effects, such as [4, 5, 7, 13, 44, 45]. In contrast to these previous works focusing on long-term causal effects, this paper aims to balance multiple short-term and long-term causal effects.

**Trustworthy policy learning.** Trustworthy policy learning ensures that the learned policies or models are reliable and dependable for practical applications. Traditional policy learning aims to identify individuals who would maximize the utility function based on their features if treated [46]. Recently, trustworthy policy learning has focused on ensuring that the learned policy adheres to principles such as beneficence, non-maleficence, autonomy, justice, no-harm, and explicability [42, 47–49]. Various counterfactual-based metrics have been suggested to assess a policy's trustworthiness [42, 50–53]. In this paper, we complement this series of work by developing a principled policy learning approach that can effectively balance multiple rewards.

**Multi-objective optimization (MOP).** MOP aims to find compromises or trade-offs among multiple possibly contrasting objectives. It is widely used in the field of machine learning such as multi-task learning [40, 54], neural architecture search [55], and multi-objective reinforcement learning [56–58]. We extend these works to a new setting by learning the optimal policy for balancing multiple long-term and short-term rewards. Additionally, we provide a practical method for interpreting and selecting preference vectors with theoretical guarantees.

## 6   Conclusion

In this paper, we focus on learning the optimal policy for balancing multiple long-term and short-term rewards. We reveal the limitations of the previous linear weighting method, which usually results in sub-optimal policies in practice. To address these limitations, we formulate the policy learning problem as a multi-objective optimization problem and then propose the novel DPPL method to learn optimal policies. The DPPL method obtains a set of Pareto optimal policies by solving a series of subproblems based on pre-specified preference vectors, effectively balancing multiple objectives. Furthermore, we theoretically establish the connection between the optimization subproblems in the DPPL method and the $\varepsilon$-constraint problem. This connection aids decision-makers in better understanding and selecting preference vectors. We conducted extensive experiments on two benchmark datasets which validate the effectiveness of our proposed method. A limitation of this work is that it focuses on discrete treatments in identification and estimation (Section 2.3). In some application scenarios, continuous treatments (e.g., price) are of interest. Further investigation is required to extend the proposed method to accommodate such cases.

## Acknowledgements

Qinwei Yang, Xueqing Liu, Yan Zeng, and Peng Wu were supported by the National Natural Science Foundation of China (No. 12301370, 62473009), the funding from the Beijing Municipal Education Commission for the Emerging Interdisciplinary Platform for Digital Business at Beijing Technology and Business University, the Beijing Key Laboratory of Applied Statistics and Digital Regulation, and the gift funding from ByteDance Research.

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

# A Estimation of Short-Term and Long-Term Rewards

For a given policy $\pi(\boldsymbol{\theta})$, the policy values for the short-term outcome $S_i$ and the long-term outcome $Y_j$ are defined as

$$\mathcal{V}(\boldsymbol{\theta}; s_i) = \mathbb{E}[\pi(\boldsymbol{\theta})S_i(1) + (1 - \pi(\boldsymbol{\theta}))S_i(0)], \quad i = 1, ..., I$$
$$\mathcal{V}(\boldsymbol{\theta}; y_j) = \mathbb{E}[\pi(\boldsymbol{\theta})Y_j(1) + (1 - \pi(\boldsymbol{\theta}))Y_j(0)], \quad j = 1, ..., J,$$

Under Assumptions 1-2, the short-term reward $\mathcal{V}(\boldsymbol{\theta}; s_i)$ and long-term reward $\mathcal{V}(\boldsymbol{\theta}; y_j)$ are identified as

$$\mathcal{V}(\boldsymbol{\theta}; s_i) = \mathbb{E}[\pi(\boldsymbol{\theta})\mu_{i1}(X) + (1 - \pi(\boldsymbol{\theta}))\mu_{i0}(X)],$$
$$\mathcal{V}(\boldsymbol{\theta}; y_j) = \mathbb{E}[\pi(\boldsymbol{\theta}\tilde{m}_{j1}(X, S) + (1 - \pi(\boldsymbol{\theta}))\tilde{m}_{j0}(X, S)].$$

where $\mu_{ia}(\boldsymbol{X}) = \mathbb{E}[S_i | \boldsymbol{X}, A = a]$, $\tilde{m}_{ja}(\boldsymbol{X}, \boldsymbol{S}) = \mathbb{E}[Y_j | \boldsymbol{X}, \boldsymbol{S}, A = a, R_j = 1]$ for $a = 0, 1$. The identifiability results are derived using a similar approach to that outlined in Section 5 of [8]. In addition, for estimating the $\mathcal{V}(\boldsymbol{\theta}; s_i)$ and $\mathcal{V}(\boldsymbol{\theta}; y_j)$, [8] proved the efficient bounds of $\mathcal{V}(\boldsymbol{\theta}; s_i)$ and $\mathcal{V}(\boldsymbol{\theta}; y_j)$, which we list them below for the sake of self-containedness.

**Lemma A.1** (Efficiency Bounds of $\mathcal{V}(\boldsymbol{\theta}; s_i)$ and $\mathcal{V}(\boldsymbol{\theta}; y_j)$, [8]). *Let* $\mathbf{Z} = (\boldsymbol{X}, A, \boldsymbol{S}, \boldsymbol{Y})$, *under Assumptions 1-2, we have that*

*(a) the efficient influence function of $\mathcal{V}(\boldsymbol{\theta}; s_i)$ is $\phi_{s_i} - \mathcal{V}(\boldsymbol{\theta}; s_i)$, where*

$$\begin{aligned}
\phi_{s_i} &= \phi_{s_i}(\mathbf{Z}; e, \mu_{i0}, \mu_{i1}) \\
&= \{\pi(\boldsymbol{\theta})\mu_{i1}(\boldsymbol{X}) + (1 - \pi(\boldsymbol{\theta}))\mu_{i0}(\boldsymbol{X})\} \\
&\quad + \frac{\pi(\boldsymbol{\theta})A(S_i - \mu_{i1}(\boldsymbol{X}))}{e(\boldsymbol{X})} + \frac{(1 - \pi(\boldsymbol{\theta}))(1 - A)(S_i - \mu_{i0}(\boldsymbol{X}))}{1 - e(\boldsymbol{X})},
\end{aligned}$$

*and $e(\boldsymbol{X}) = \mathbb{P}(A = 1 | \boldsymbol{X})$ is propensity score. The associated semiparametric efficiency bound of $\mathcal{V}(\boldsymbol{\theta}; s_i)$ is Var($\phi_{s_i}$).*

*(b) the efficient influence function of $\mathcal{V}(\boldsymbol{\theta}; y_j)$ is $\phi_{y_j} - \mathcal{V}(\boldsymbol{\theta}; y_j)$, where*

$$\begin{aligned}
\phi_{y_j} &= \phi_{y_j}(\mathbf{Z}; e, r_j, m_{j0}, m_{j1}, \tilde{m}_{j0}, \tilde{m}_{j1}) \\
&= \{\pi(\boldsymbol{\theta})m_{j1}(\boldsymbol{X}) + (1 - \pi(\boldsymbol{\theta}))m_{j0}(\boldsymbol{X})\} \\
&\quad + \frac{\pi(\boldsymbol{\theta})AR_j(Y_j - \tilde{m}_{j1}(\boldsymbol{X}, \boldsymbol{S}))}{e(\boldsymbol{X})r_j(1, \boldsymbol{X}, \boldsymbol{S})} + \frac{\pi(\boldsymbol{\theta})A(\tilde{m}_{j1}(\boldsymbol{X}, \boldsymbol{S}) - m_{j1}(\boldsymbol{X}))}{e(\boldsymbol{X})} \\
&\quad + \frac{(1 - \pi(\boldsymbol{\theta}))(1 - A)R_j(Y_j - \tilde{m}_{j0}(\boldsymbol{X}, \boldsymbol{S}))}{(1 - e(\boldsymbol{X}))r_j(0, \boldsymbol{X}, \boldsymbol{S})} \\
&\quad + \frac{(1 - \pi(\boldsymbol{\theta}))(1 - A)(\tilde{m}_{j0}(\boldsymbol{X}, \boldsymbol{S}) - m_{j0}(\boldsymbol{X}))}{1 - e(\boldsymbol{X})},
\end{aligned}$$

$m_{ja}(\boldsymbol{X}) = \mathbb{E}[Y_j | \boldsymbol{X}, A = a, R_j = 1]$ *is the regression function for $Y_j$, and $r_j(A, \boldsymbol{X}, \boldsymbol{S}) = \mathbb{P}[R_j = 1 | \boldsymbol{X}, \boldsymbol{S}, A]$ is selection score. The associated semiparametric efficiency bound of $\mathcal{V}(\boldsymbol{\theta}; y_j)$ is Var($\phi_{y_j}$).*

From Lemma A.1, for a given policy $\pi(\boldsymbol{\theta})$, it is natural to define the estimators of $\mathcal{V}(\boldsymbol{\theta}; s_i)$ and $\mathcal{V}(\boldsymbol{\theta}; y_j)$ as

$$\hat{\mathcal{V}}(\boldsymbol{\theta}; s_i) = \frac{1}{N} \sum_{n=1}^{N} \phi_{s_i}(Z_n; \hat{e}, \hat{\mu}_{i0}, \hat{\mu}_{i1}),$$

$$\hat{\mathcal{V}}(\boldsymbol{\theta}; y_j) = \frac{1}{N} \sum_{n=1}^{N} \phi_{y_j}(Z_n; \hat{e}, \hat{r}_j, \hat{m}_{j0}, \hat{m}_{j1}, \hat{\tilde{m}}_{j0}, \hat{\tilde{m}}_{j1}).$$

where $N$ is the sample size. All of them can be identified from the observed data. And $\hat{e}(\boldsymbol{x}), \hat{\mu}_{ia}(\boldsymbol{x}), \hat{m}_{ja}(\boldsymbol{x}), \hat{\tilde{m}}_{ja}(\boldsymbol{x}, \boldsymbol{s})$, and $\hat{r}_j(a, \boldsymbol{x}, \boldsymbol{s})$ for $a = 0, 1$ are the estimators of $e(\boldsymbol{x}), \mu_{ia}(\boldsymbol{x}), m_{ja}(\boldsymbol{x}), \tilde{m}_{ja}(\boldsymbol{x}, \boldsymbol{s})$ and $r_j(a, \boldsymbol{x}, \boldsymbol{s})$ respectively.

# B  Algorithm Flowchart for DPPL

---
**Algorithm 1** DPPL Algorithm

---
1: **Input:** A set of preference vectors $\{\mathbf{u}_1, \mathbf{u}_2, ..., \mathbf{u}_K\}$
   (All subproblems can be solved in parallel)
2: **for** $k = 1$ to $K$ **do**
3:    randomly generate parameters $\boldsymbol{\theta}_r^{(k)}$
4:    find the initial parameters $\boldsymbol{\theta}_0^{(k)}$ from $\boldsymbol{\theta}_r^{(k)}$ using gradient-based method (step a)
5:    **for** $t = 1$ to $T$ **do**
6:       obtain $\lambda_{tm}^{(k)} \geq 0, \beta_{tk'}^{(k)} \geq 0, \forall m = 1, ..., M, \forall k' \in I_\epsilon(\boldsymbol{\theta})$ by solving subproblem (6)
7:       calculate direction $\boldsymbol{d}_t^{(k)} = -\left(\sum_{i=m}^M \lambda_{tm}^{(k)} \nabla \bar{\mathcal{V}}_m(\boldsymbol{\theta}_t^{(k)}) + \sum_{k' \in I_{\epsilon(\theta)}} \beta_{tk'}^{(k)} \nabla \mathcal{G}_{k'}(\boldsymbol{\theta}_t^{(k)})\right) / \boldsymbol{d}_t^{(k)} = $
         $-(\lambda_{tm}^k + \sum_{k' \in I_{\epsilon(\theta)}} \beta_{tk'}^k (\boldsymbol{u}_{k'm} - \boldsymbol{u}_{km})) \nabla \bar{\mathcal{V}}_m(\boldsymbol{\theta}_t^k)$   (step b)
8:       update the parameters $\boldsymbol{\theta}_{t+1}^{(k)} = \boldsymbol{\theta}_t^{(k)} + \eta d_t^{(k)}$
9:    **end for**
10: **end for**
11: **Output:** The set of solutions for all subproblems with different trade-offs $\{\boldsymbol{\theta}_T^{(k)} | k = 1, \ldots, K\}$

---

# C  Proofs of Theorem 1

For the case of only have one long-term outcome $Y$ and one short-term outcome $S$, considering the $\varepsilon$-constraint optimization problem

$$\min_{\boldsymbol{\theta}} \bar{\mathcal{V}}(\boldsymbol{\theta}; y), \quad s.t., \bar{\mathcal{V}}(\boldsymbol{\theta}; s) \leq \varepsilon \tag{A.1}$$

and the linear weighting optimization problem

$$\min_{\boldsymbol{\theta}} \omega_1 \bar{\mathcal{V}}(\boldsymbol{\theta}; y) + \omega_2 \bar{\mathcal{V}}(\boldsymbol{\theta}; s) \tag{A.2}$$

which can be reformulated as

$$\min_{\boldsymbol{\theta}} \bar{\mathcal{V}}(\boldsymbol{\theta}; y) + \lambda \bar{\mathcal{V}}(\boldsymbol{\theta}; s).$$

where $\lambda = \omega_2/\omega_1$ controls the balance between short-term and long-term rewards. Let $\tau_s(\boldsymbol{X}) = \mathbb{E}[S(1) - S(0)|\boldsymbol{X}]$ and $\tau_y(\boldsymbol{X}) = \mathbb{E}[Y(1) - Y(0)|\boldsymbol{X}]$. When $\lambda = 0$, it is equivalent to finding an optimal policy for minimizing $\bar{\mathcal{V}}(\boldsymbol{\theta}; y)$ alone, $\pi_y^*(\boldsymbol{\theta}) = \arg\min_\pi \bar{\mathcal{V}}(\boldsymbol{\theta}; y) = \arg\min_\pi -\mathbb{E}[\pi(\boldsymbol{\theta})\tau_y(\boldsymbol{X})] = \mathbb{I}(\tau_y(\boldsymbol{X}) \geq 0)$. When $\lambda = \infty$, it is equivalent to finding an optimal policy for minimizing the $\bar{\mathcal{V}}(\boldsymbol{\theta}; s)$ alone, $\pi_s^*(\boldsymbol{\theta}) = \arg\min_\pi \bar{\mathcal{V}}(\boldsymbol{\theta}; s) = \arg\min_\pi -\mathbb{E}[\pi(\boldsymbol{\theta})\tau_s(\boldsymbol{X})] = \mathbb{I}(\tau_s(\boldsymbol{X}) \geq 0)$. We have the following theorem:

**Theorem C.1.** *For the weights $\boldsymbol{\omega}$ in problem* (A.5)*, and the thresholds $\varepsilon$ in problem* (A.1)*, the following statements hold:*

- *When $\varepsilon < -\mathbb{E}[\pi_s^*(\boldsymbol{\theta})S(1) + (1 - \pi_s^*(\boldsymbol{\theta}))S(0)]$, the solution of the constrained optimization problem is empty.*

- *When $\varepsilon \geq -\mathbb{E}[\pi_s^*(\boldsymbol{\theta})S(1) + (1 - \pi_s^*(\boldsymbol{\theta}))S(0)]$, the relationship between $\lambda$ and $\alpha$ is described as follows:*

  - *$\lambda = 0$, if $\varepsilon \geq -\mathbb{E}[\pi_y^*(\boldsymbol{\theta})S(1) + (1 - \pi_y^*(\boldsymbol{\theta}))S(0)]$.*
  - *$\lambda$ is the solution of the equation*

$$-\mathbb{E}[\mathbb{I}(\tau_y(\boldsymbol{X}) + \lambda\tau_s(\boldsymbol{X}) > 0) \cdot \tau_s(\boldsymbol{X}) + \mu_0(\boldsymbol{X})] = \varepsilon,$$

  *if $-\mathbb{E}[\pi_s^*(\boldsymbol{\theta})S(1) + (1 - \pi_s^*(\boldsymbol{\theta}))S(0)] < \varepsilon \leq -\mathbb{E}[\pi_y^*(\boldsymbol{\theta})S(1) + (1 - \pi_y^*(\boldsymbol{\theta}))S(0)]$.*

It is important to note that for a given $\lambda$, we could solve the value of $\varepsilon$ by solving the equation

$$-\mathbb{E}[\mathbb{I}(\tau_y(\boldsymbol{X}) + \lambda\tau_s(\boldsymbol{X}) > 0) \cdot \tau_s(\boldsymbol{X}) + \mu_0(\boldsymbol{X})] = \varepsilon,$$

as the left side of the equation is a monotone function of $\lambda$ and the solution is unique, and all the quantities such as $\tau_s(\boldsymbol{X}), \tau_y(\boldsymbol{X})$, and $\mu_0(\boldsymbol{X})$ are identifiable.

*Proof.* Initially, we recognize that $\varepsilon$ cannot be too small so that no policy can satisfy the constraint of $\bar{\mathcal{V}}(\boldsymbol{\theta}; s) \le \varepsilon$. The optimal policy of minimizing only the $\bar{\mathcal{V}}(\boldsymbol{\theta}; s)$ is $\pi_s^*(\boldsymbol{\theta}) = \mathbb{I}(\tau_s(\boldsymbol{X}) \ge 0)$. Thus, $\varepsilon \ge -\mathbb{E}[\pi_s^*(\boldsymbol{\theta})S(1) + (1 - \pi_s^*(\boldsymbol{\theta}))S(0)]$.

When $\varepsilon \ge -\mathbb{E}[\pi_s^*(\boldsymbol{\theta})S(1) + (1 - \pi_s^*(\boldsymbol{\theta}))S(0)]$. First, the optimal policy of minimizing only $\bar{\mathcal{V}}(\pi; y)$ is given as $\pi_y^*(\boldsymbol{\theta}) = \mathbb{I}(\tau_y(\boldsymbol{X}) \ge 0)$. Then, $\varepsilon \le -\mathbb{E}[\pi_y^*(\boldsymbol{\theta})S(1) + (1 - \pi_y^*(\boldsymbol{\theta}))S(0)]$. otherwise, the constraint will be invalid and the constrained optimization problem becomes an unconstrained optimization problem with $\lambda = 0$.

Second, when $-\mathbb{E}[\pi_s^*(\boldsymbol{\theta})S(1) + (1 - \pi_s^*(\boldsymbol{\theta}))S(0)] \le \varepsilon \le -\mathbb{E}[\pi_y^*(\boldsymbol{\theta})S(1) + (1 - \pi_y^*(\boldsymbol{\theta}))S(0)]$, we show that the optimal policy $\pi^*(\boldsymbol{\theta})$ parameterized by $\boldsymbol{\theta}^*$, for the constrained optimization problem

$$\min_{\boldsymbol{\theta}} \bar{\mathcal{V}}(\boldsymbol{\theta}; y), \quad s.t., \bar{\mathcal{V}}(\boldsymbol{\theta}; s) \le \varepsilon$$

is obtained only when $\bar{\mathcal{V}}(\boldsymbol{\theta}^*; s) = \varepsilon$. Below, we prove it with the method of reduction to absurdity. If $-\mathbb{E}[\pi_s^*(\boldsymbol{\theta})S(1) + (1 - \pi_s^*(\boldsymbol{\theta}))S(0)] \le \varepsilon \le -\mathbb{E}[\pi_y^*(\boldsymbol{\theta})S(1) + (1 - \pi_y^*(\boldsymbol{\theta}))S(0)]$, then there are some units that satisfies $\{\tau_s(\boldsymbol{X}) < 0, \tau_y(\boldsymbol{X}) > 0\}$ that not being assigned treatment by $\pi^*(\boldsymbol{\theta})$; otherwise, the constraint $\bar{\mathcal{V}}(\boldsymbol{\theta}; s) \le \varepsilon$ will be violated. Thus, we could find another treatment policy $\tilde{\pi}^*$ that assigns more treatment to the units with $\{\tau_s(\boldsymbol{X}) < 0, \tau_y(\boldsymbol{X}) > 0\}$, which yields a lower $\bar{\mathcal{V}}(\boldsymbol{\theta}; y)$ but increases the $\bar{\mathcal{V}}(\boldsymbol{\theta}; s)$. That is, $\tilde{\pi}^*$ will lead to a $\bar{\mathcal{V}}(\boldsymbol{\theta}; s)$ closer to $\varepsilon$ but has a lower $\bar{\mathcal{V}}(\boldsymbol{\theta}; y)$ than $\pi^*$, thus, $\pi^*$ is not the optimal policy, which contradicts its definition of $\pi^*$. Thus, the constrained optimization problem becomes

$$\min_{\boldsymbol{\theta}} \bar{\mathcal{V}}(\boldsymbol{\theta}; y), \quad s.t., \bar{\mathcal{V}}(\boldsymbol{\theta}; s) = \varepsilon.$$

By introducing the Lagrange multiplier $\beta$, $\pi^*$ satisfies

$$\pi^* = \arg\min_{\pi} \quad \bar{\mathcal{V}}(\boldsymbol{\theta}; y) + \beta\bar{\mathcal{V}}(\boldsymbol{\theta}; s) = \mathbb{I}(\tau_y(\boldsymbol{X}) + \beta\tau_s(\boldsymbol{X}) > 0),$$

where $\beta$ is the solution of $\bar{\mathcal{V}}(\boldsymbol{\theta}^*; s) = \varepsilon$, i.e.,

$$-\mathbb{E}[\mathbb{I}(\tau_y(\boldsymbol{X}) + \beta\tau_s(\boldsymbol{X}) > 0)\tau_s(\boldsymbol{X}) + \mu_0(\boldsymbol{X})] = \varepsilon.$$

This completes the proof for Theorem C.1. $\qquad\square$

We can further extend Theorem C.1 to situations where there are multiple long-term rewards and multiple short-term rewards. More generally, for the $\varepsilon$-constraint optimization problem

$$\min_{\boldsymbol{\theta}} \bar{\mathcal{V}}_l(\boldsymbol{\theta}), \text{ s. t. } \bar{\mathcal{V}}_m(\boldsymbol{\theta}) \le \varepsilon_m \text{ for all } m = 1, \ldots, M, m \ne l, \tag{A.3}$$

and the linear weighting optimization problem

$$\min_{\boldsymbol{\theta}} \bar{\mathcal{V}}(\boldsymbol{\theta}) = \sum_{i=m}^{M} \omega_m \bar{\mathcal{V}}_m(\boldsymbol{\theta}), \tag{A.4}$$

where $\omega_m$ is the pre-specified weight for the $m$-th reward. We have the following theorem:

**Theorem 1.** *For the preference vector $\boldsymbol{u}_k$ in problem* (1)*, the weights $\boldsymbol{\omega}$ in problem* (2)*, and the thresholds $\boldsymbol{\varepsilon}$ in problem* (8)*, the following statements hold:*

*(a) the connection between $\boldsymbol{\varepsilon}$ and $\boldsymbol{\omega}$ is given as*

$$-\mathbb{E}[\mathbb{I}(\tau_l(\boldsymbol{X}) + \frac{\omega_m}{\omega_l}\tau_m(\boldsymbol{X}) > 0) \cdot \tau_m(\boldsymbol{X}) + h_m(\boldsymbol{X})] = \varepsilon_m, \text{ for } m = 1 \cdots M, \text{ and } m \ne l,$$

*where $\tau_m(\boldsymbol{X})$ is the conditional average causal effects for $m$-th short/long-term outcome,*

$$\tau_m(\boldsymbol{X}) = \begin{cases} \mathbb{E}[S_i(1) - S_i(0)|\boldsymbol{X}], & \text{if } \omega_m \text{ is the weight of } \bar{\mathcal{V}}(\boldsymbol{\theta}, s_i), \\ \mathbb{E}[Y_j(1) - Y_j(0)|\boldsymbol{X}], & \text{if } \omega_m \text{ is the weight of } \bar{\mathcal{V}}(\boldsymbol{\theta}, y_j), \end{cases}$$

*and*

$$h_m(\boldsymbol{X}) = \begin{cases} \mathbb{E}[S_i(0)|\boldsymbol{X}], & \text{if } \omega_m \text{ is the weight of } \bar{\mathcal{V}}(\boldsymbol{\theta}, s_i), \\ \mathbb{E}[Y_j(0)|\boldsymbol{X}, \boldsymbol{S}, R_j = 1], & \text{if } \omega_m \text{ is the weight of } \bar{\mathcal{V}}(\boldsymbol{\theta}, y_j), \end{cases}$$

*and $\mathbb{I}(\cdot)$ is the indicator function.*

*(b) the connection between $\boldsymbol{\omega}$ and $\boldsymbol{u}_k$ is given as*

$$\omega_m = \lambda_m + \sum_{k' \in \mathcal{I}_\epsilon(\boldsymbol{\theta})} \beta_{k'}(\boldsymbol{u}_{k'm} - \boldsymbol{u}_{km}), \; for \; m = 1, \cdots, M,$$

*where $\lambda_m$ and $\beta_{k'}$ are defined in Eq.(6), $\mathcal{I}_\epsilon(\boldsymbol{\theta}) = \{k' | \mathcal{G}_{k'}(\boldsymbol{\theta}) \geq -\epsilon\}$*

*Proof.* First, for the Theorem1(a), combining the TheoremC.1, more generally, for the $\varepsilon$-constraint problem

$$\min_{\boldsymbol{\theta}} \; \bar{\mathcal{V}}_l(\boldsymbol{\theta}), \; \text{s. t. } \bar{\mathcal{V}}_m(\boldsymbol{\theta}) \leq \varepsilon_m \text{ for all } m = 1, \ldots, M, m \neq l, \tag{A.5}$$

and the linear weighting optimization problem

$$\min_{\boldsymbol{\theta}} \bar{\mathcal{V}}(\boldsymbol{\theta}) = \sum_{i=m}^{M} \omega_m \bar{\mathcal{V}}_m(\boldsymbol{\theta}), \tag{A.6}$$

where $\omega_m$ is the pre-specified weight for the $m$-th reward. By mathematical induction, we have:

$$-\mathbb{E}[\mathbb{I}(\tau_l(\boldsymbol{X}) + \omega_m/\omega_l \tau_m(\boldsymbol{X}) > 0)\tau_m(\boldsymbol{X}) + h_{m0}(\boldsymbol{X})] = \varepsilon_i, m = 1 \cdots M, and \; m \neq l$$

where $\tau_m(\boldsymbol{X}) = \begin{cases} \tau_{s_i} = \mathbb{E}[S_i(1) - S_i(0)|\boldsymbol{X}], & \text{if } \omega_m \text{ is the weight of } \bar{\mathcal{V}}(\boldsymbol{\theta}, s_i), \\ \tau_{y_j} = \mathbb{E}[Y_j(1) - Y_j(0)|\boldsymbol{X}], & \text{if } \omega_m \text{ is the weight of } \bar{\mathcal{V}}(\boldsymbol{\theta}, y_j), \end{cases}$

$h_{m0}(X) = \begin{cases} \mu_{i0}(\boldsymbol{X}) = \mathbb{E}[S_i|\boldsymbol{X}, A = 0], & \text{if } \omega_m \text{ is the weight of } \bar{\mathcal{V}}(\boldsymbol{\theta}, s_i), \\ \tilde{m}_{j0}(\boldsymbol{X}, \boldsymbol{S}) = \mathbb{E}[Y_j|\boldsymbol{X}, \boldsymbol{S}, A = a, R_j = 1], & \text{if } \omega_m \text{ is the weight of } \bar{\mathcal{V}}(\boldsymbol{\theta}, y_j), \end{cases}$

This completes the proof for Theorem 1(a)

Second, for the Theorem (b), motivated by [40], for constraint problem

$$\begin{aligned} (\boldsymbol{d}_t, \alpha_t) = \arg \min_{d \in \mathbb{R}^n, \alpha \in \mathbb{R}} \; & \alpha + \frac{1}{2}||\boldsymbol{d}||^2 \\ s.t. \quad & \nabla \bar{\mathcal{V}}_m(\boldsymbol{\theta}_t)^T \boldsymbol{d} \leq \alpha, m = 1, ..., M. \\ & \nabla \mathcal{G}_{k'}(\boldsymbol{\theta}_t)^T \boldsymbol{d} \leq \alpha, k' \in \mathcal{I}_\epsilon(\boldsymbol{\theta}_t), \end{aligned} \tag{A.7}$$

we have

$$\nabla \mathcal{G}_{k'}(\theta_t) = (\boldsymbol{u}_{k'} - \boldsymbol{u}_k)^T \nabla \bar{\mathcal{V}}(\boldsymbol{\theta}_t) = \sum_{m=1}^{M} (\boldsymbol{u}_{k'm} - \boldsymbol{u}_{km}) \nabla \bar{\mathcal{V}}_m(\boldsymbol{\theta}_t). \tag{A.8}$$

Base on KKT conditions, we have

$$d_t = -(\sum_{m=1}^{M} \lambda_m \nabla \bar{\mathcal{V}}_m(\boldsymbol{\theta}_t) + \sum_{k' \in I_\epsilon(\boldsymbol{\theta})} \beta_{k'} \nabla \mathcal{G}_{k'}(\boldsymbol{\theta}_t)), \sum_{m=1}^{M} \lambda_m + \sum_{k' \in I_\epsilon(\boldsymbol{\theta})} \beta_{k'} = 1, \tag{A.9}$$

where $\lambda_m \leq 0$ and $\beta_{k'} \leq 0$ are the Lagrange multipliers. Then, the dual problem is given as

$$\max_{\lambda_m, \beta_{k'}} -\frac{1}{2}|| \sum_{m=1}^{M} \lambda_m \nabla \bar{\mathcal{V}}_m(\boldsymbol{\theta}_t) + \sum_{k' \in I_\epsilon(\boldsymbol{\theta})} \beta_{k'} \nabla \mathcal{G}_{k'}(\boldsymbol{\theta}_t)||^2$$

$$s.t. \quad \sum_{m=1}^{M} \lambda_m + \sum_{k' \in I_\epsilon(\boldsymbol{\theta})} \beta_{k'} = 1, \lambda_m \geq 0, \beta_{k'} \geq 0, \forall m = 1, ..., M, \forall \; k' \in \mathcal{I}_\epsilon(\boldsymbol{\theta}). \tag{A.10}$$

Substituting Eq.(A.8) into Eq.A.9, we have

$$d_t = -\left(\sum_{m=1}^{M} \lambda_m \nabla \bar{\mathcal{V}}_m(\boldsymbol{\theta}_t) + \sum_{k' \in I_\epsilon(\boldsymbol{\theta})} \beta_{k'} \left(\sum_{m=1}^{M} (\boldsymbol{u}_{k'm} - \boldsymbol{u}_{km}) \nabla \bar{\mathcal{V}}_m(\boldsymbol{\theta}_t)\right)\right)$$

$$= -\left(\lambda_m + \sum_{k' \in I_{\epsilon(\theta)}} \beta_{k'} (\boldsymbol{u}_{k'm} - \boldsymbol{u}_{km})\right) \nabla \bar{\mathcal{V}}_m(\boldsymbol{\theta}_t) \qquad \text{(A.11)}$$

For the problem(A.6), $d_t$ is the negative gradient direction. Thus, we have

$$\bar{\mathcal{V}}(\boldsymbol{\theta}) = \sum_{m=1}^{M} \omega_m \bar{\mathcal{V}}_m(\boldsymbol{\theta}), \text{where } \omega_m = \lambda_m + \sum_{k' \in I_{\epsilon(\theta)}} \beta_{k'} (\boldsymbol{u}_{k'm} - \boldsymbol{u}_{km}), \qquad \text{(A.12)}$$

where $\lambda_m$ and $\beta_{k'}$ is obtained from Eq.(A.10). This shows that the DPPL method can be transformed into the linear weighting method. This completes the proof for Theorem 1(b) □

## D  Additional Experimental Results

### D.1  Sensitivity Analysis on Missing Ratio

In the following, we show more experimental result with missing ratio $r = 0.3$ under IHDP and JOBS datasets, in table D1.

In additional, We show the corresponding $\varepsilon$ value for each preference vector with different missing ratio $\{0.3, 0.4, 0.5\}$ under IHDP and JOBS datasets, in tables D2, D3 and D7.

Table D1: Comparison of our method (OURS) and linear weighting method (LW) with 10 preference vectors on IHDP and JOBS, with Short-Term Reward (S-REWARDS) and Long-Term Reward (L-REWARDS), $\Delta W$ and Variance (S-VAR and L-VAR) as evaluation metrics. The missing ratio $r = 0.3$ and $T = 4$. The best result is bolded.

| IHDP | S-REWARDS | | L-REWARDS | | $\Delta W$ | | S-VAR | | L-VAR | |
|---|---|---|---|---|---|---|---|---|---|---|
| PREFERENCE VECTOR | OURS | LW | OURS | LW | OURS | LW | OURS | LW | OURS | LW |
| 1 (1.00, 0.00) | **523.060** | 520.760 | **389.485** | 385.990 | **41.174** | 38.277 | **12.673** | 13.621 | **49.054** | 58.344 |
| 2 (0.98, 0.17) | **526.880** | 524.900 | 376.918 | **377.275** | **36.801** | 35.989 | 15.593 | **12.336** | **60.458** | 63.531 |
| 3 (0.94, 0.34) | 522.300 | **522.440** | 386.931 | **393.534** | 39.517 | **42.889** | 14.998 | **12.181** | **51.183** | 59.204 |
| 4 (0.86, 0.50) | 522.280 | **523.300** | **376.800** | 376.515 | **34.642** | 34.559 | **12.591** | 15.040 | 61.746 | **51.064** |
| 5 (0.76, 0.64) | **523.480** | 518.440 | 380.358 | **398.327** | 36.820 | **43.285** | **12.959** | 15.250 | 59.013 | **47.264** |
| 6 (0.64, 0.76) | **525.560** | 517.800 | 387.263 | **390.716** | **41.313** | 39.160 | **13.703** | 14.991 | 56.639 | **49.135** |
| 7 (0.50, 0.87) | **523.420** | 517.440 | **389.624** | 385.813 | **41.424** | 36.528 | **12.594** | 18.091 | **59.317** | 59.628 |
| 8 (0.34, 0.94) | **521.880** | 515.280 | 383.755 | **388.985** | **35.719** | 35.034 | **12.690** | 14.945 | **48.189** | 57.030 |
| 9 (0.17, 0.98) | **520.300** | 515.800 | 386.994 | **399.012** | 38.549 | **42.308** | **13.622** | 19.584 | 56.273 | 58.640 |
| 10 (0.00, 1.00) | **522.500** | 514.980 | 381.171 | **385.961** | **36.737** | 35.372 | **12.455** | 19.711 | **43.415** | 49.718 |

| JOBS | S-REWARDS | | L-REWARDS | | $\Delta W$ | | S-VAR | | L-VAR | |
|---|---|---|---|---|---|---|---|---|---|---|
| PREFERENCE VECTOR | OURS | LW | OURS | LW | OURS | LW | OURS | LW | OURS | LW |
| 1 (1.00, 0.00) | **1615.540** | 1612.100 | **1221.629** | 1217.543 | **155.936** | 152.173 | 65.386 | **56.393** | 98.666 | **92.897** |
| 2 (0.98, 0.17) | **1616.240** | 1600.280 | 1216.370 | **1217.547** | **153.657** | 146.265 | **58.903** | 75.467 | **87.611** | 92.085 |
| 3 (0.94, 0.34) | **1616.380** | 1595.840 | **1229.393** | 1219.475 | **160.238** | 145.009 | **57.370** | 86.875 | 95.219 | **91.009** |
| 4 (0.86, 0.50) | **1615.700** | 1592.200 | **1234.526** | 1201.847 | **162.465** | 134.375 | **56.556** | 88.052 | **89.535** | 94.647 |
| 5 (0.76, 0.64) | **1608.600** | 1595.260 | 1214.387 | **1219.359** | **148.846** | 144.661 | **57.526** | 95.379 | **79.273** | 99.852 |
| 6 (0.64, 0.76) | **1612.120** | 1591.480 | **1222.689** | 1221.671 | **154.756** | 143.927 | **55.446** | 97.238 | **94.283** | 97.522 |
| 7 (0.50, 0.87) | **1614.240** | 1588.660 | **1225.527** | 1220.786 | **157.235** | 142.075 | **58.574** | 104.776 | **85.414** | 108.986 |
| 8 (0.34, 0.94) | **1607.880** | 1585.280 | **1227.527** | 1223.203 | **155.055** | 141.593 | **55.923** | 105.193 | **85.365** | 101.940 |
| 9 (0.17, 0.98) | **1610.600** | 1584.460 | 1221.183 | **1223.446** | **153.243** | 141.305 | **59.996** | 109.731 | **92.968** | 99.344 |
| 10 (0.00, 1.00) | **1612.740** | 1590.880 | 1211.837 | **1224.826** | **149.640** | 145.205 | **60.330** | 106.403 | **92.767** | 106.106 |

Table D2: The $\varepsilon$ values corresponding to each preference vector in the two datasets IHDP and JOBS, where $T = 4$ and $r = 0.3$, which are derived according to Theorem 1.

| PREFERENCEVECTOR | IHDP $-\varepsilon$ | JOBS $-\varepsilon$ |
|---|---|---|
| (1.00, 0.00) | 0.827 | 0.865 |
| (0.98, 0.17) | 0.818 | 0.864 |
| (0.94, 0.34) | 0.825 | 0.859 |
| (0.86, 0.50) | 0.830 | 0.858 |
| (0.77, 0.64) | 0.800 | 0.858 |
| (0.64, 0.76) | 0.722 | 0.841 |
| (0.50, 0.86) | 0.592 | 0.738 |
| (0.34, 0.94) | 0.549 | 0.706 |
| (0.17, 0.98) | 0.539 | 0.726 |
| (0.00, 1.00) | 0.557 | 0.779 |

Table D3: The $\varepsilon$ values corresponding to each preference vector in the two datasets IHDP and JOBS, where $T = 4$ and $r = 0.4$, which are derived according to Theorem 1.

| PREFERENCEVECTOR | IHDP $-\varepsilon$ | JOBS $-\varepsilon$ |
|---|---|---|
| (1.00, 0.00) | 0.822 | 0.877 |
| (0.98, 0.17) | 0.824 | 0.868 |
| (0.94, 0.34) | 0.823 | 0.852 |
| (0.86, 0.50) | 0.820 | 0.841 |
| (0.77, 0.64) | 0.813 | 0.806 |
| (0.64, 0.76) | 0.724 | 0.798 |
| (0.50, 0.86) | 0.524 | 0.703 |
| (0.34, 0.94) | 0.512 | 0.694 |
| (0.17, 0.98) | 0.523 | 0.667 |
| (0.00, 1.00) | 0.523 | 0.666 |

Table D4: The $\varepsilon$ values corresponding to the preference vectors in the two datasets IHDP and JOBS, where $T = 4$ and $r = 0.5$, which are derived according to Theorem 1.

| PREFERENCEVECTOR | IHDP $-\varepsilon$ | JOBS $-\varepsilon$ |
|---|---|---|
| (1.00, 0.00) | 0.820 | 0.865 |
| (0.98, 0.17) | 0.826 | 0.863 |
| (0.94, 0.34) | 0.821 | 0.869 |
| (0.86, 0.50) | 0.816 | 0.853 |
| (0.77, 0.64) | 0.805 | 0.816 |
| (0.64, 0.76) | 0.679 | 0.781 |
| (0.50, 0.86) | 0.522 | 0.737 |
| (0.34, 0.94) | 0.489 | 0.722 |
| (0.17, 0.98) | 0.490 | 0.723 |
| (0.00, 1.00) | 0.541 | 0.684 |

## D.2 Sensitivity Analysis on Preference Vector

In the following, we show more experimental result with different numbers of preference vectors $K = \{4, 8, 12\}$ under JOBS datasets, in table D5-D7.

Table D5: Comparison of our method (OURS) and linear weighting method (LW) with 4 preference vectors on JOBS, with Short-Term Reward (S-REWARDS) and Long-Term Reward (L-REWARDS), $\Delta W$ and Variance (S-VAR and L-VAR) as evaluation metrics. The missing ratio $r = 0.2$ and $T = 4$. The best result is bolded.

| JOBS | S-REWARDS | | L-REWARDS | | $\Delta W$ | | S-VAR | | L-VAR | |
|---|---|---|---|---|---|---|---|---|---|---|
| PREFERENCE VECTOR | OURS | LW | OURS | LW | OURS | LW | OURS | LW | OURS | LW |
| 1(1.00, 0.00) | **1616.540** | 1613.940 | 1226.493 | **1232.147** | 158.869 | **160.396** | 60.171 | **57.758** | 94.783 | **92.298** |
| 2(0.87, 0.50) | **1606.920** | 1599.620 | **1226.861** | 1222.933 | **154.242** | 148.628 | **60.608** | 77.760 | **78.282** | 92.699 |
| 3(0.50, 0.86) | **1612.500** | 1601.260 | **1226.470** | 1213.741 | **156.837** | 144.852 | **58.438** | 82.862 | **87.381** | 94.363 |
| 4(0.00, 1.00) | **1615.740** | 1596.360 | **1224.834** | 1223.110 | **157.639** | 147.087 | **58.856** | 86.287 | **86.425** | 87.150 |

Table D6: Comparison of our method (OURS) and linear weighting method (LW) with 8 preference vectors on JOBS, with Short-Term Reward (S-REWARDS) and Long-Term Reward (L-REWARDS), $\Delta W$ and Variance (S-VAR and L-VAR) as evaluation metrics. The missing ratio $r = 0.2$ and $T = 4$. The best result is bolded.

| JOBS | S-REWARDS | | L-REWARDS | | $\Delta W$ | | S-VAR | | L-VAR | |
|---|---|---|---|---|---|---|---|---|---|---|
| PREFERENCE VECTOR | OURS | LW | OURS | LW | OURS | LW | OURS | LW | OURS | LW |
| 1(1.00, 0.00) | **1616.340** | 1615.940 | **1233.387** | 1227.531 | **162.215** | 159.088 | **55.283** | 57.953 | **93.105** | 95.263 |
| 2(0.97, 0.22) | **1610.820** | 1605.220 | **1228.604** | 1223.384 | **157.064** | 151.654 | **63.065** | 66.878 | **85.019** | 87.506 |
| 3(0.90, 0.43) | **1606.260** | 1599.940 | 1212.864 | **1226.675** | 146.914 | **150.659** | **60.809** | 70.123 | **95.023** | 97.185 |
| 4(0.78, 0.62) | **1614.960** | 1604.000 | **1226.162** | 1222.282 | **157.913** | 150.493 | **62.883** | 78.589 | 94.868 | **89.057** |
| 5(0.62, 0.78) | **1612.320** | 1594.220 | **1226.816** | 1225.956 | **156.920** | 147.440 | **59.959** | 77.331 | **81.906** | 89.562 |
| 6(0.43, 0.90) | **1611.860** | 1593.840 | 1221.652 | **1225.291** | **154.108** | 141.917 | **60.059** | 82.969 | 99.027 | **90.661** |
| 7(0.22, 0.97) | **1612.700** | 1596.060 | 1215.533 | **1224.358** | **151.468** | 147.561 | **56.015** | 89.441 | **86.439** | 92.825 |
| 8(0.00, 1.00) | **1612.580** | 1592.260 | **1233.756** | 1227.061 | **160.520** | 147.012 | **58.113** | 83.188 | **88.058** | 98.805 |

Table D7: Comparison of our method (OURS) and linear weighting method (LW) with 12 preference vectors on JOBS, with Short-Term Reward (S-REWARDS) and Long-Term Reward (L-REWARDS), $\Delta W$ and Variance (S-VAR and L-VAR) as evaluation metrics. The missing ratio $r = 0.2$ and $T = 4$. The best result is bolded.

| JOBS | S-REWARDS | | L-REWARDS | | $\Delta W$ | | S-VAR | | L-VAR | |
|---|---|---|---|---|---|---|---|---|---|---|
| PREFERENCE VECTOR | OURS | LW | OURS | LW | OURS | LW | OURS | LW | OURS | LW |
| 1(1.00, 0.00) | 1610.800 | **1614.600** | 1231.786 | **1232.158** | 158.645 | **160.731** | **56.774** | 60.101 | 89.746 | **87.940** |
| 2(0.98, 0.14) | 1609.720 | **1610.740** | **1224.605** | 1222.904 | **154.515** | 154.174 | **59.048** | 60.887 | **88.027** | 92.283 |
| 3(0.95, 0.28) | **1613.520** | 1606.320 | **1228.204** | 1226.660 | **158.214** | 153.842 | **59.249** | 65.024 | 84.400 | **79.787** |
| 4(0.91, 0.41) | **1615.600** | 1598.940 | 1223.297 | **1231.718** | **156.800** | 152.681 | **58.106** | 70.854 | 98.610 | **88.081** |
| 5(0.84, 0.54) | **1614.140** | 1604.860 | 1218.585 | **1220.647** | **153.714** | 150.105 | **61.420** | 65.414 | **89.712** | 95.134 |
| 6(0.75, 0.65) | **1615.240** | 1598.960 | **1227.954** | 1225.251 | **158.949** | 149.457 | **54.882** | 76.213 | **86.061** | 89.256 |
| 7(0.65, 0.75) | **1616.380** | 1596.160 | **1226.506** | 1218.845 | **158.795** | 144.854 | **61.503** | 77.284 | **91.857** | 95.620 |
| 8(0.54, 0.84) | **1613.420** | 1598.460 | 1223.097 | **1229.293** | **155.610** | 151.228 | **58.566** | 83.100 | **92.342** | 97.554 |
| 9(0.41, 0.91) | **1612.940** | 1594.320 | 1222.586 | **1224.040** | **155.115** | 146.532 | **57.262** | 84.387 | **87.325** | 93.589 |
| 10(0.28, 0.95) | **1612.980** | 1596.880 | **1230.538** | 1218.671 | **159.111** | 145.127 | **61.465** | 81.949 | **92.381** | 95.530 |
| 11(0.14, 0.98) | **1612.160** | 1591.760 | 1214.424 | **1224.345** | **150.644** | 145.404 | **58.148** | 86.692 | **80.732** | 90.234 |
| 12(0.00,1.00) | **1613.040** | 1592.440 | **1228.213** | 1224.288 | **157.978** | 145.716 | **63.826** | 87.624 | **84.520** | 87.628 |

