# OpenReview forum: "Learning the Optimal Policy for Balancing Short-Term and Long-Term Rewards"
_NeurIPS.cc/2024/Conference — NeurIPS 2024 poster_

### Official Review · Reviewer_B23s · 2024-07-09

**Soundness:** 3
**Presentation:** 3
**Contribution:** 3
**Rating:** 6
**Confidence:** 3

**Summary:**

This paper introduces a new way to balance multiple rewards with some long-term rewards potentially missing. It does so by using Pareto Policy Learning of optimizing each reward subject up to the tradeoff frontier. This can be more practical than simple linear weighting since the linear weighting strategy applies the constant weight regardless of the amount of conflict between pairs of objectives. Empirically, the papers show that the approach is superior to linear on two synthetic tasks with some real data. Overall I think the paper is promising and adding more realistic empirical evaluation can add values to the current state of the paper.

**Strengths:**

- Learning to combine multiple rewards is an important and well-motivated question, and has wide ranging implications.
- The method proposed is mathematically sound. The paper shows theoretically that the input parameters can be interpreted as a form of worst case value on each objective.
- The paper explains how the most popular approach of linear weighting can fall short, derives the method through first principles, and empirically demonstrates that the proposed method is superior.

**Weaknesses:**

- The main weakness of the paper is that the experimentation is rather limited. The experiment uses partial real data with synthetic generation of short-term and long-term rewards. For example, in robotic planning, the authors could show how their approach helps balance the long-term reward (e.g. goal reaching) / short-term reward (e.g. minimizing jerk). This is just an example, but including other more real-world planning and RL problems would seem beneficial.
- It seems that compared to linear weighting, the proposed method seeks more short-term reward but is not necessarily better in terms of long-term reward. It may not be a weakness but reading the table does strike me that the method is more “short-sighted.”

**Questions:**

- The separation between short-term and long-term reward is practically meaningful, but mathematically the only difference is that one reward can be missing and the other is fully observable, since the Pareto Policy Learning treats all objectives the same. Do we necessarily need these separate definitions? Can we put everything as a long-term reward that can be missing sometimes?
- How are the weighting and preference vectors chosen? Have the author considered running a sweep over different configurations to compare against linear weighting? Apologies if I overlook this detail from the paper.

**Limitations:**

- The limitation of the overall framework are mentioned but there is no much detail perhaps due to space constraint.

---

> ### Author Rebuttal · Authors · 2024-08-05
>
> Thank you for approving our work and for the helpful suggestions. Below, we address your concerns and questions.
>
> >**W1**: The experiment uses partial real data with synthetic generation of short-term and long-term rewards.
>
> **Response:** Thanks for your comments. We fully agree that applying the proposed method to other domains, such as robotic planning and RL, would be beneficial.
>
> However, **we would like to clarify that our method focuses on policy learning in causal inference, which has peculiarities that cannot be correctly evaluated using real-world data**. Here are the reasons:
>
> In causal inference, the primary focus is on estimating treatment effects. Various methods achieve this using observed data under several identifiability assumptions. However, **it is impossible to evaluate a method's performance based solely on observed data because the true treatment effects are always unknown in real-world data. Therefore, almost all experiments in causal inference [1-5] are conducted using synthetic data where the true treatment effects are known.**
>
> **Next, we give a detailed explanation.** Consider a binary treatment $A\in\{0,1\}$, covariates $X$, and outcome $Y$. Although $X$, $A$, and $Y$ are observed variables, they cannot alone define treatment effects. To define treatment effects, we denote $Y(a)$ as the potential outcome if $A$ were set to $a$ ($a$=0, 1). Thus, each individual has two potential outcomes: $Y(1)$ and $Y(0)$. The observed outcome $Y_i = A_i Y_i(1) + (1-A_i) Y_i(0)$, i.e., the observed outcome is the potential outcome corresponding to the received treatment. Then individualized treatment effect is defined as
> $$ ITE_i=Y_i(1)-Y_i(0),$$
> which measures the treatment effect for individual $i$. **Since each individual can only receive one treatment, we observe either $Y_i(0)$ or $Y_i(1)$, but not both. This is the fundamental problem of causal inference. Thus, we cannot obtain the true ITE from the observed data because we cannot simultaneously observe $Y_i(0)$ and $Y_i(1)$.** In this article, we study the policy learning problem, which is also a causal problem. Consistent with the above analysis, due to the fundamental problem of causal inference, the rewards for a given policy cannot be evaluated using observed data alone.
>
> > **W2:** It seems that compared to linear weighting, the proposed method seeks more short-term reward but is not necessarily better in terms of long-term reward.
>
> **Response:** Thanks for your insightful comments. In this article, **we aim to develop a policy learning approach that balances both short- and long-term rewards, rather than focusing solely on maximizing either of them.**
>
> To evaluate the proposed method, we mainly use three metrics: S-REWARD, L-REWARD, and $\Delta W$. S-REWARD measures the short-term reward induced by the learned policy,  L-REWARD measures the long-term reward, and $\Delta W$ measures the balanced welfare, indicating the policy's ability to trade off between multiple rewards. Clearly, **to evaluate an approach that balances both short-term and long-term reward, $\Delta W$ is the most important metric.**
>
> In the experimental results, the proposed method may not always perform better on  L-REWARD, but it shows superior performance for $\Delta W$. **This indicates that the linear weighting approach does not achieve a good balance, as it pursues long-term rewards at the expense of short-term rewards. In contrast, our approach proves to be Pareto optimal, pursuing long-term rewards while also maximizing short-term rewards.**
>
> > **Q1**: Do we necessarily need these separate definitions? Can we put everything as a long-term reward that can be missing sometimes?
>
> **Response:**  Thank you for raising this point. Yes, both long-term and short-term outcomes can be missing, and the proposed policy learning method remains applicable. However, this will bring extra difficulty in identifying and estimating long and short-term rewards. From a high-level perspective, the proposed policy learning approach consists of two steps:
>
> * Step 1: policy evaluation, estimating the short and long-term rewards;
> * Step 2: policy learning, solving optimization problem (3) based on the estimated rewards.
>
> The missingness of short and long-term outcomes mainly affects Step 1, which involves identifying and estimating the rewards. Once the rewards are estimated, Step 2 focuses on learning the optimal policy through advanced optimization algorithms.
>
> In Step 1, when both outcomes are missing, the estimation method of [6] may be invalid. Developing novel methods for identifying and estimating rewards in this context is an interesting direction, and we leave it for future work.
>
> > **Q2**: How are the weighting and preference vectors chosen?
>
> **Response:**  Thanks for your comments. Preference vectors (PVs) are used to quantify an individual's preference for different objectives. **In our article, we randomly generate 10 unit PVs (see lines 270-272), ensuring the consistency and comparability of the preference measures.** Each component of the PV represents the importance of the decision maker's preference for different objectives. Each PV is considered as a weight vector in the linear weighting method, see lines 275-276.  **In addition, we added experiments for different number of PVs, see response W5 to reviewer hVvL.**
>
> [1] Johansson et al. Learning representations for counterfactual inference, ICML 2016
>
> [2] Shalit et al. Estimating individual treatment effect: generalization bounds and algorithms.ICML 2017
>
> [3] Shi et al. Adapting neural networks for the estimation of treatment effects. NeurIPS 2019
>
> [4] Yoon et al. GANITE: Estimation of individualized treatment effects using generative adversarial nets, ICLR 2018
>
> [5] Bica et al. Estimating the effects of continuous-valued interventions using generative adversarial networks. NeurIPS 2020
>
> [6] Wu et al. Policy Learning for Balancing Short-Term and Long-Term Rewards. arXiv 2024

---

> > ### Author Response · Authors · 2024-08-13
> > **Kindly Reminder**
> >
> > Dear Reviewer B23s,
> >
> > We are deeply grateful for the time and effort you have invested in reviewing our paper. We have endeavoured to address your questions as thoroughly as possible. As the discussion period draws to a close, we kindly inquire if there are any further concerns or questions that we might assist with.
> >
> > Sincerely,
> >
> > Authors #17045

---

### Official Review · Reviewer_hVvL · 2024-07-09

**Soundness:** 3
**Presentation:** 3
**Contribution:** 3
**Rating:** 4
**Confidence:** 2

**Summary:**

This paper attempts to address the challenge of learning the optimal policy for balancing multiple long-term and short-term rewards. The authors point out that the existing linear weighting method leads to a sub-optimal policy. To address this limitation, the authors propose formulating formulate the problem as a multi-objective optimization problem. They utilize the Lagrange algorithm to use preference vectors to solve the formulated multi-objective optimization problem and aim to learn the policy to meet Pareto optimization. In order to decide the preference vectors, the authors propose establishing the connection between the optimization problems and the ε-constraint problem. Experiments on IHDP and JOBS demonstrate the efficacy of the proposed method.

**Strengths:**

1.	The multi-object problem is practical in both reinforcement learning and other optimization scenarios. The paper provides a good summary of the limitations of the existing linear weighting method and introduces a novel perspective on solving the problem by resorting to the Lagrange algorithm and Pareto optimization.
2.	The author has a solid mathematical foundation and is able to provide detailed mathematical descriptions and solutions to the proposed optimization problems.

**Weaknesses:**

1. The authors point out that the linear weighting method is suboptimal. However, there is no explanation in the method section or corresponding experiments to demonstrate that the proposed method (i.e. DPPL) is optimal.

2. In line 38, the authors claim that when some of the rewards are interrelated, the linear weighting method can only achieve a suboptimal solution. The claim may not be rigorous as the linear weighting method might be able to model the relationship among the rewards. More explanation and experiments are required.

3. In line 95, the definition of Pareto optimality, the condition for Pareto optimality by the author is to find the $\theta$ that makes all $\bar{\mathcal{V}}$ optimal. However, is it possible that the $\theta$ is not optimal for some $\bar{\mathcal{V}}$ but is optimal for the overall $\bar{\mathcal{V}}$?

4. Some mathematical symbols and proprietary terms in the paper are not explained clearly. For example, what does the $e$ in line 110 mean? What does MOP represent? Does MOP represent multi-objective problems? What do $v$ and $R_{+}$ mean in line 171? What does the KKT condition mean? Is it the KKT condition in the Lagrange algorithm? What is the difference between the two descent directions $d_{rt}$ and $d_t$? There are many similar situations in the paper. I suggest providing necessary explanations for each noun and symbol that appears for the first time.

5. In section Simulating Output and section Experimental Details, many parameters are defined by the authors themselves, but most of them do not have reasons or ablation experiments. For example, why is the number of preference vectors 10? In Line 253 to Line 254, why are some parameters truncated normal distributions and some Gaussian distributions?

6. In Table 1 on the L-REWARDS metric, the proposed method is comparable to the linear weighting method. However, the authors claim that for most of the preference vectors, DPPL's solutions have better performance.

7. In Figure 1, it seems that the effect on the $\delta{w}$ from the missing rate and T is not obvious for either the proposed method or LW. More explanation is needed.

**Questions:**

Please see the comments above.

**Limitations:**

Yes

---

> ### Author Rebuttal · Authors · 2024-08-05
>
> Below, we hope to address your concerns and questions.
>
> >**W1** No enough explanation/experiments to demonstrate that the proposed method is optimal.
>
> > **W2**: When some of the rewards are interrelated, the linear weighting method can only achieve a suboptimal solution. The claim may not be rigorous. More explanation and experiments are required.
>
> **Response:** Thanks for your comments. Since W1 and W2 are similar, we response them together.
>
> **First, we explain why the proposed method is optimal. In Lemma 2 of Section 3.2 (lines 191-194), we show that the proposed method achieves Pareto optimality.** Specifically, a Pareto optimal solution is obtained by solving the optimization problem (5) (lines 181-182) using an iterative gradient-based update rule. Lemma 2 demonstrates that this iterative rule converges to the Pareto optimal solutions.
>
> **Second, we explain why the linear weighting (LW) method is suboptimal**.
>  - Previous studies [1-4] highlight the limitations of LW method in multi-objective optimization. LW method is restricted by assumptions of linearity and convexity, potentially ignoring complex interactions among objectives.
> - When multiple objectives have nonlinear relationships, the Pareto set (the collection of all Pareto optimal solutions, see line 100) may not be convex. Due to its linear restriction, the LW method may not effectively identify Pareto optimal solutions within non-convex regions, leading to suboptimal solutions.
> - As illustrated in Figure 2 of [5], when dealing with two objective functions, the LW method may struggle to identify Pareto-optimal solutions when the Pareto set is nonconvex.
>
> **Finally, we conducted an extra experiment to further illustrate the limitations of LW method and the optimality of our method.** We consider two objectives
> $min_x F(x)=[f_1(x),f_2(x)]^T,$
> $$f_1(x)=1-\exp{(-\sum_{i=1}^d(x_d-\frac1{\sqrt{d}})^2)}, f_2(x)=1-\exp{(-\sum_{i=1}^d(x_d+\frac1{\sqrt{d}})^2)},$$
> where $x$ is a 20-dimensional covariates and $x_d$ is the $d$-th element of $x$. For the optimization problem above, we know the true Pareto set. For clarity, we present 10 uniformly selected Pareto solutions in Table 1 below, from which we can see that the Pareto set is concave rather than convex.
>
> **Table 1**
> |id|f1_pf|f2_pf|
> |-|-|-|
> |0|0.982|0.000|
> |1|0.960|0.041|
> |2|0.921|0.153|
> |3|0.854|0.313|
> |4|0.754|0.486|
> |5|0.617|0.647|
> |6|0.452|0.777|
> |7|0.279|0.870|
> |8|0.126|0.930|
> |9|0.026|0.966|
>
> Next, we use both the proposed method and the LW method to solve the above optimization problem. Here we select ten preference vectors (they also are the weight vectors in the LW method). The results are shown in Table 2 below,  where $(f_{1, LW},f_{2, LW})$ and $(f_{1, our},f_{2,our})$ represents the solution obtained by the LW method and our method, respectively. **we can see that the LW method only finds solutions at the endpoints of the Pareto set (i.e., in the convex part), while our method successfully locates all Pareto optimal solutions within the whole region.** This indicates that the LW method cannot achieve the Pareto optimal solutions.
>
> **Table 2**
> |PreferenceVector|$f_{1,LW}$|$f_{2,LW}$||$f_{1,our}$|$f_{2,our}$|
> |-|-|-|-|-|-|
> |(1.00,0.00)|0.002|0.980||0.953|0.060|
> |(0.98,0.17)|0.000|0.981||0.918|0.160|
> |(0.94,0.34)|0.000|0.981||0.823|0.374|
> |(0.86,0.50)|0.001|0.980||0.736|0.511|
> |(0.76,0.64)|0.001|0.979||0.632|0.632|
> |(0.64,0.76)|0.979|0.001||0.632|0.632|
> |(0.50,0.87)|0.007|0.974||0.511|0.736|
> |(0.34,0.94)|0.981|0.000||0.334|0.844|
> |(0.17,0.98)|0.981|0.000||0.228|0.892|
> |(0.00,1.00)|0.979|0.002||0.055|0.955|
>
> > **W3**: For Pareto optimality, is it possible that $\theta$ is not optimal for some $\mathcal{\bar V}$ but is optimal for the overall $\mathcal{\bar V}$?
>
> **Response:** Thanks for your comments. We kindly remind the reviewer that there might be a misunderstanding regarding the definition of Pareto optimality. Below, we provide a detailed explanation. A solution is Pareto optimal if it is impossible to improve on objective without worsening other objectives. **Typically, multi-objective problems have numerous Pareto optimal solutions, and all of them form the Pareto set. These solutions represent the best trade-offs among different objectives, rather than optimality for a specific single objective.** Thus, it is possible that a Pareto optimal solution achieves the optimal solution in some objectives but not for others.
>
> > **W4**: Some mathematical symbols and proprietary terms in the paper are not explained clearly.
>
> **Response:** Thanks for your comments and we apologize for any lack of clarity. Below, we provide a detailed explanation of the mathematical symbols:
> - We define $e(x) \triangleq P(A=1|X =x)$ (see line 110).
> - The MOP represents multi-objective optimization (see line 91).
> - In line 171, $v$ is an $M$-dimensional vector in the positive real space $R_+^M$, where $M$ is the number of objectives.
> - The Karush-Kuhn-Tucker (KKT) conditions are a set of necessary conditions that characterize the solutions to constrained optimization problems. The KKT conditions can be seen as a set of constraints derived from the Lagrangian function that must be met for a solution to be considered optimal in a constrained optimization problem.
> - Both $d_{r_t}$ and $d_{t}$ represent the descent direction at $t$-th iteration, where the former is for seeking the initial solution, while the latter is for finding the Pareto optimal solution.
>
> ---
> [1] Miettinen. Nonlinear multiobjective optimization, 2012
>
> [2] Marler et al. The weighted sum method for multi-objective optimization: new insights, 2010
>
> [3] Censor. Pareto optimality in multiobjective problems,1977
>
> [4] Ngatchou et al. Pareto multi objective optimization, 2005
>
> [5] Shim et al. Pareto‐based continuous evolutionary algorithms for multiobjective optimization, 2002
>
> (Regarding the response to W5-W7, please see global review.)

---

> > ### Author Response · Authors · 2024-08-13
> > **Kindly Reminder**
> >
> > Dear Reviewer hVvL,
> >
> > We are deeply grateful for the time and effort you have invested in reviewing our paper. We have endeavoured to address your questions as thoroughly as possible. As the discussion period draws to a close, we kindly inquire if there are any further concerns or questions that we might assist with.
> >
> > Sincerely,
> >
> > Authors #17045

---

### Official Review · Reviewer_xruw · 2024-07-12

**Soundness:** 3
**Presentation:** 3
**Contribution:** 3
**Rating:** 5
**Confidence:** 1

**Summary:**

This paper studies the tradeoff between short-term and long-term rewards. The authors formulate the policy learning problem as a multi-objective optimization problem and propose a decomposition-based Pareto policy learning method. I only had experience in reinforcement learning in robotics five years ago. I tried my best to understand the paper, but I am not sure about my rating and comments.

**Strengths:**

- This paper studies a quite interesting and important problem, and the proposed methods seem effective on these two benchmarks.
- The paper is well-organized, the division is relatively easy to follow, and the proposed method is well-motivated.

**Weaknesses:**

- Only the linear weighting method is used as the baseline. I am wondering if there are any other methods that can be used for comparison. If not, why? Since both IHDP and JOBS are widely used.

**Questions:**

Please see the weakness.

**Limitations:**

Yes

---

> ### Author Rebuttal · Authors · 2024-08-05
>
> We sincerely appreciate your comments and thank you for the helpful suggestions. Below, we hope to address your concerns and questions.
>
> > **W1**: - Only the linear weighting method is used as the baseline. I am wondering if there are any other methods that can be used for comparison. If not, why? Since both IHDP and JOBS are widely used.
>
> **Response:**  Thanks for your comments. From a high-level perspective, the proposed policy learning approach consists of the following two steps:
>
> * Step 1: policy evaluation, estimating the short-term and long-term rewards $\mathcal{V}(\theta, s_i)$ and $\mathcal{V}(\theta, y_j)$;
>
> * Step 2: policy learning, solving the optimization problem (3) based on the estimated values of $\mathcal{V}(\theta, s_i)$ and $\mathcal{V}(\theta, y_j)$.
>
> For Step 1, the previous work [1] is well-established. However, for Step 2, it only uses a simple linear weighting method. In this article, we primarily focus on Step 2 and adopt the method of [1] in Step 1, given in Appendix A.
>
> **As discussed in the second paragraph of the Introduction (lines 32-35), the policy learning problem of balancing short and long-term rewards remains largely unexplored. There is only a single work [1] that addresses this problem, using a simple linear weighting method. Therefore, our work primarily compares with this approach.**
>
> In addition, we also tried to consider the epsilon constraint method, but it can be converted into a linear weighted method through certain transformations, so we did not choose it as an additional baseline.
>
> Nevertheless, to make a more comprehensive comparison, we added two extra estimation methods in Step 1, and compare our proposed optimization method with the linear weighting method in Step 2.  Specifically, the two extra estimators are given as
>
> * OR estimator
> $$\hat{\mathbb{V}}(\pi;s)^{OR}=\frac{1}{n}\sum_{i=1}^{n} [\pi(X_{i}) \hat{\mu}_{1}(X_i)+(1-\pi(X_i))\hat{\mu}_0(X_i)]$$
>
> $$\hat{\mathbb{V}}(\pi;y)^{OR}=\frac{1}{n}\sum_{i=1}^{n}[\pi(X_{i}) \hat{\tilde{m}}_1(X_i,S_i)+(1-\pi(X_i)) \hat{\tilde{m}}_0(X_i,S_i)]$$
>
> * DR estimator
> $$\hat{\mathbb{V}}(\pi;s)^{DR}=\frac{1}{n}\sum_{i=1}^{n}\Big[\pi(X_i)\left(\frac{A_i(S_i-\hat{\mu}_1(X_i)}{\hat{e}(X_i)}+\hat{\mu}_1(X_i)\right)+(1-\pi(X_i))\left(\frac{(1-A_i)(S_i-\hat{\mu}_0(X_i)}{1-\hat{e}(X_i)}+\hat{\mu}_0(X_i)\right)\Big]$$
>
> $$\hat{\mathbb{V}}(\pi;y)^{DR}=\frac{1}{n}\sum_{i=1}^{n}\Big[\pi(X_i)\left(\frac{A_i(Y_i-\hat{\tilde{m}}_1(X_i,S_i)}{\hat{e}(X_i)}+\hat{\tilde{m}}_1(X_i,S_i)\right)+(1-\pi(X_i))\left(\frac{(1-A_i)(Y_i-\hat{\tilde{m}}_0(X_i,S_i)}{1-\hat{e}(X_i)}+\hat{\tilde{m}}_0(X_i,S_i)\right)\Big]$$
> The associated experimental results on JOBS are presented below.
>
> | JOBS| S-Rewards | | L-Rewards | | $\Delta{W}$ | |  S-VAR | |  L-VAR  | |
> |:-----------------:|:---------:|:--------:|:---------:|:--------:|:---------:|:-------:|:------:|:-------:|:-------:|:------:|
> | Preference Vector | OURS | LW | OURS | LW | OURS | LW |OURS | LW | OURS | LW |
> | (1.00, 0.00)| 1612.70 | **1614.44** | **1226.81** | 1212.77 | **157.11** | 150.96 | 60.53 | **60.12** | 102.07 | **87.59**  |
> | (0.98, 0.17) | **1611.94** | 1604.74 | **1220.29** | 1216.57 | **153.47** | 148.01 | **60.19** | 65.98 | 100.92 | **93.77** |
> | (0.94, 0.34) | **1613.46** | 1597.26 | **1220.38** | 1215.91 | **154.27** | 143.94 | **62.48** | 77.57 | **86.22** | 87.80 |
> | (0.86, 0.50) | **1614.34** | 1595.88 | **1228.16**| 1218.13 | **158.60** | 144.36 | **58.28** | 82.76 | 91.15 | **90.15** |
> |(0.76, 0.64)| **1616.20** | 1597.76 | 1219.02 | **1219.98** | **154.96**  | 146.22 | **56.09** | 88.02 | **90.36** | 92.34 |
> | (0.64, 0.76)|**1614.58** | 1592.96 | **1219.77** | 1217.16 | **154.53**  | 142.41 | **56.96** | 92.09 | **86.43** | 90.10 |
> | (0.50, 0.87)| **1614.64** | 1586.78 | 1212.08| **1223.47** | **150.71**  | 142.48 | **60.22** | 104.24 | **88.82** | 96.13 |
> | (0.34, 0.94) | **1611.14** | 1587.86 | 1212.31 | **1222.62** |**149.08** | 142.59 | **52.62** | 109.01 | **91.18** | 91.67 |
> | (0.17, 0.98) | **1612.24** | 1580.14 | 1222.36 | **1226.82** |**154.65** | 140.83 | **54.33** | 109.42 | **87.33** | 95.10 |
> | (0.00, 1.00) | **1613.48** | 1586.04 | 1222.03 | **1227.29** |**155.11** | 144.02 | **60.04** | 107.85 | **87.18** | 90.78 |
>
>
> | JOBS | S-Rewards |    | L-Rewards |    | $\Delta{W}$ |  |  S-VAR |    |  L-VAR |    |
> |:---------------:|:---------:|:--------:|:---------:|:--------:|:---------:|:-------:|:------:|:-------:|:------:|:------:|
> | Preference Vector |OURS| LW |OURS| LW |OURS |LW | OURS | LW | OURS | LW |
> |(1.00, 0.00)|1607.82|**1612.26**|**1230.10**|1224.90| **156.31**|155.93| 63.43| **59.50**|90.06| **87.42**|
> |(0.98, 0.17)|**1616.08**|1598.50|**1230.45**|1223.66| **160.61** |148.43|**58.27**|65.18| 95.92| **90.49**|
> |(0.94, 0.34)| **1612.72**|1598.14|1218.20|**1218.87**| **152.81** |145.86|**57.97**|80.05|85.47| **84.29**|
> |(0.86, 0.50)|**1610.58**|1592.74|1219.66|**1224.71**| **152.47** |146.08|**57.60**|84.19|**80.43**| 95.71|
> |(0.76, 0.64)|**1608.50**|1594.00|**1228.59**|1221.82| **155.90** |145.26|**57.27**|89.49|**81.03** | 97.67  |
> |(0.64, 0.76)|**1616.34**|1592.82|1229.27|**1234.92**| **160.16** |151.22|**58.65**|87.72| **90.82** | 95.56  |
> |(0.50, 0.87)|**1612.54**|1593.30|1217.99|**1227.72**| **152.62** |147.86|**57.13**|101.56| **88.11** | 96.09  |
> |(0.34, 0.94)|**1613.38**|1584.02|**1224.66**|1220.44| **156.37** |139.58|**58.84**|104.60| **90.69** | 93.02  |
> |(0.17, 0.98)|**1612.54**|1591.02|**1228.40**|1224.59| **157.82** |145.16|**56.88**|110.24| 83.45  | **82.22**  |
> |(0.00, 1.00)|**1613.74**|1588.70|1218.97|**1237.01**| **153.71** |150.21|**58.38**|106.48| **90.94**  | 99.12  |
>
>
> From the experimental results, we can see that our method achieves better performance on the evaluation metric $\Delta{W}$ (the most important evaluation metric), and our method also performs better in terms of overall performance.
>
>
> [1] Wu et al. Policy Learning for Balancing Short-Term and Long-Term Rewards. arXiv:2405.03329, 2024.

---

> > ### Author Response · Authors · 2024-08-13
> > **Kindly Reminder**
> >
> > Dear Reviewer xruw,
> >
> > We are deeply grateful for the time and effort you have invested in reviewing our paper. We have endeavoured to address your questions as thoroughly as possible. As the discussion period draws to a close, we kindly inquire if there are any further concerns or questions that we might assist with.
> >
> > Sincerely,
> >
> > Authors #17045

---

### Official Review · Reviewer_ouKL · 2024-07-12

**Soundness:** 3
**Presentation:** 3
**Contribution:** 3
**Rating:** 6
**Confidence:** 3

**Summary:**

This paper proposes a framework for solving multi-objective optimization problems: multi-objective optimization problems are divided into sub-problems in different regions by setting different preference vectors. The parameter optimization direction of the sub-problem can be easily solved by transforming it into a dual problem through the KKT condition, and a Pareto optimal solution of the original problem can be obtained by solving the sub-problem. This paper uses this framework to balance the optimal strategy learning under multiple short-term rewards and long-term rewards and achieves better and more stable performance than the traditional linear weighted method in the constructed experimental environment.

**Strengths:**

1. This paper reveals in detail the connection between the proposed method and the linear weighted method and the epsilon-constrained optimization method. Based on this connection, the epsilon-constrained optimization method can provide interpretability for the method in this paper.
2. The method in this paper theoretically overcomes the suboptimality problem of the linear weighted method and avoids the situation where the epsilon-constrained optimization method does not have a feasible solution.
3. This paper obtains better and more stable results than the epsilon-constrained optimization method in the optimal strategy learning problem under multiple short-term rewards and long-term rewards constructed by the author.

**Weaknesses:**

1. This paper mainly proposes an important multi-objective optimization algorithm and compares it with two existing algorithms in theory. However, the title of this paper seems to be just a specific application scenario of the algorithm. In what other scenarios can this algorithm be applied?
2. The experimental part is mainly conducted in a constructed environment, and it is unclear how difficult it is in the field of causal inference.
3. The v in line 171 is missing \bar. In Appendix B, t in line 5 of Algorithm 1 should start from 0.

**Questions:**

1. How to deal with a situation where long-term rewards are missing? Should the data be ignored when solving the network?
I think the optimization method proposed in this paper and the connection with related methods are very interesting, and I will improve my score as appropriate.

**Limitations:**

Have addressed.

---

> ### Author Rebuttal · Authors · 2024-08-05
>
> We sincerely appreciate your approval of the idea and the novelty of this work and thank you for the helpful suggestions. Below, we hope to address your concerns and questions.
>
> > **W1**:  This paper proposes an important multi-objective optimization algorithm. But the title of this paper seems to be just a specific application scenario. In what other scenarios can this algorithm be applied?
>
> **Response:**  Thank you for the insightful comments. As the reviewer noted, we propose a general multi-objective optimization algorithm, making the proposed method applicable to various scenarios involving multi-objective balance.  Balancing multiple objectives is particularly critical in the field of trustworthy AI, where the goal is not only to achieve desirable prediction performance but also to address other important aspects such as fairness [1,2] and non-harm [3,4].
>
> > **W2**: The experimental part is mainly conducted in a constructed environment, and it is unclear how difficult it is in the field of causal inference.
>
> **Response:**  Thanks for the constructive comments. In the field of causal inference, the primary focus is on estimating  treatment effects. With several identifiability assumptions, various methods have been developed to achieve this goal using the observed data only. **However, it is impossible to evaluate a method's performance based solely on the observed data. The basic rationale is that the true treatment effects are always unknown in real-world data.** Thus, the experiments in almost all the articles in causal inference [5-9] are conducted in a constructed environment where the true treatment effects are known.
>
> **Next, we give a detailed explanation.**  Consider the case of a binary treatment variable $A\in\{0,1\}$, $X$ is covariates, and $Y$ is the outcome. Although $X$, $A$, and $Y$ are all observed variables, they cannot alone define treatment effects. To define treatment effects, we denote $Y(a)$ as the potential outcome that would be observed if $A$ were set to $a$ ($a$=0, 1). Thus, each individual has two potential outcomes: one is $Y(1)$ if the individual receives the treatment ($A=1$), and the other is $Y(0)$ if the individual don't receives the treatment ($A=0$). The observed outcome $Y_i = A_i Y_i(1) + (1-A_i) Y_i(0)$, that is,  the observed outcome is the potential outcome corresponding to the received treatment. With the notation of potential outcomes,  we define the individualized treatment effect as
> $$ ITE_i=Y_i(1)-Y_i(0),$$
> which measures the magnitude of the treatment effect for individual $i$. **Since each individual can only receive one treatment, we observe either $Y_i(0)$ or $Y_i(1)$, but not both. This  is known as the fundamental problem of causal inference [10]. As a result, we cannot obtain the true ITE from the observed data because we cannot simultaneously observe $Y_i(0)$ and $Y_i(1)$.**
>
> In this article, we aim to find the optimal policy that maximizes rewards $V(\pi)=E[\pi(X)Y(1)+(1-\pi(X))Y(0)]$, which is also a causal problem. Consistent with the above analysis, due to the fundamental problem of causal inference, for a given policy $\pi$, the rewards cannot be evaluated with the observed data only.
>
> > **W3**: The v in line 171 is missing \bar. In Appendix B, t in line 5 of Algorithm 1 should start from 0.
>
> **Response:** Thank you for pointing this out. We will revise it accordingly.
>
> > **Q1**: How to deal with a situation where long-term rewards are missing? Should the data be ignored when solving the network?
>
> **Response:**   Thank you for raising this point and we apologize for the lack of clarity. From a high-level perspective, the proposed policy learning approach consists of the following two steps:
> * Step 1: policy evaluation, estimating the short-term and long-term rewards $\mathcal{V}(\theta, s_i)$ and $\mathcal{V}(\theta, y_j)$ for a given policy;
> * Step 2: policy learning, solving the optimization problem (3)  based on the estimated values of $\mathcal{V}(\theta, s_i)$ and $\mathcal{V}(\theta, y_j)$.
>
> The missingness of long-term outcomes primarily affects Step 1, which involves the the identifiability and estimation of  $\mathcal{V}(\theta, s_i)$ and $\mathcal{V}(\theta, y_j)$ (see line 81 of the manuscript). Once $\mathcal{V}(\theta, s_i)$ and $\mathcal{V}(\theta, y_j)$ are estimated, Step 2 focuses on learning the optimal policy by developing advanced optimization algorithms.
>
> For Step 1, the previous work [11] is well-established. However, it does not address Step 2, as it only uses a simple linear weighting method.  In this article, we primarily focus on Step 2 by proposing a novel optimization algorithm. Additionally, we establish connections with other methods (e.g., $\epsilon$-constraint) to guide the selection and provide interpretation of the preference vectors. For Step 1, we adopt the method of [11] and the associated results are presented in Appendix A.
>
> **References**
>
> ---
> [1] Kusner et al. Counterfactual fairness. NeurIPS 2017
>
> [2] Ethan et al., Fairness-Oriented Learning for Optimal Individualized Treatment, JASA 2023
>
> [3] Kallus N., Treatment Effect Risk: Bounds and Inference, FAccT 2022
>
> [4] Li et al., Trustworthy Policy Learning under the Counterfactual No-Harm Criterion, ICML 2023
>
> [5] Johansson et al. Learning representations for counterfactual inference, ICML 2016
>
> [6] Shalit et al. Estimating individual treatment effect: generalization bounds and algorithms. ICML 2017
>
> [7] Shi et al. Adapting neural networks for the estimation of treatment effects. NeurIPS 2019
>
> [8] Yoon et al. GANITE: Estimation of individualized treatment effects using generative adversarial nets, ICLR 2018
>
> [9] Bica et al. Estimating the effects of continuous-valued interventions using generative adversarial networks. NeurIPS, 2020.
>
> [10] Holland. Statistics and causal inference. JASA 1986
>
> [11] Wu et al. Policy Learning for Balancing Short-Term and Long-Term Rewards. arXiv 2024

---

> > ### Author Response · Authors · 2024-08-13
> > **Kindly Reminder**
> >
> > Dear Reviewer  ouKL,
> >
> > We are deeply grateful for the time and effort you have invested in reviewing our paper. We have endeavoured to address your questions as thoroughly as possible. As the discussion period draws to a close, we kindly inquire if there are any further concerns or questions that we might assist with.
> >
> > Sincerely,
> >
> > Authors #17045

---

### Author Rebuttal · Authors · 2024-08-05

Dear Reviewer hVvL, we provide the responses to W1-W4 below your Official Review. Here, we further response W5-W7.

> **W5**: In experiment, why choose 10 preference vectors? why are some parameters truncated normal distributions.

**Response:** Thanks for your comments. **We would like to clarify that our data generation mechanism follows from previous works [6-8] for ease of comparison. In addition, we further performed two more experiments with varying numbers of preference vectors and different truncation thresholds for normal distributions.**

First, we generate $K$ unit preference vectors (the same as the article), but with $K$=4 and 12. The results, shown in Tables 3-4 below, indicates that the proposed method stably performs better.

**Table 3, $K=4$**
|JOBS|S-Rewards||L-Rewards||$\Delta{W}$||S-VAR||L-VAR||
|:-:|:-:|:-:|:-:|:-:|:-:|:-:|:-:|:-:|:-:|:-:|
|PreferenceVector|OURS|LW|OURS|LW|OURS|LW|OURS|LW|OURS|LW|
|(1.00,0.00)|**1616.5**|1613.9|1226.5|**1232.2**|158.9|**160.4**|60.2|**57.8**|94.8|**92.3**|
|(0.87,0.50)|**1606.9**|1599.6|**1226.9**|1222.9|**154.2**|148.6|**60.6**|77.8|**78.3**|92.7|
|(0.50,0.86)|**1612.5**|1601.3|**1226.5**|1213.7|**156.8**|144.9|**58.4**|82.9|**87.4**|94.4|
|(0.00,1.00)|**1615.7**|1596.4|**1224.8**|1223.1|**157.6**|147.1|**58.9**|86.3|**86.4**|87.2|

**Table 4, $K=12$**
|JOBS|S-Rewards||L-Rewards||$\Delta{W}$||S-VAR||L-VAR||
|:-:|:-:|:-:|:-:|:-:|:-:|:-:|:-:|:-:|:-:|:-:|
|PreferenceVector|OURS|LW|OURS|LW|OURS|LW|OURS|LW|OURS|LW|
|(1.00,0.00)|1610.8|**1614.6**|1231.8|**1232.2**|158.6|**160.7**|**56.8**|60.1|89.8|**87.9**|
|(0.98,0.14)|1609.7|**1610.7**|**1224.6**|1222.9|**154.5**|154.2|**59.1**|60.9|**88.0**|92.3|
|(0.95,0.28)|**1613.5**|1606.3|**1228.2**|1226.7|**158.2**|153.8|**59.3**|65.0|84.4|**79.8**|
|(0.91,0.41)|**1615.6**|1598.9|1223.3|**1231.7**|**156.8**|152.7|**58.1**|70.9|98.6|**88.1**|
|(0.84,0.54)|**1614.1**|1604.9|1218.6|**1220.7**|**153.7**|150.1|**61.4**|65.4|**89.7**|95.1|
|(0.75,0.65)|**1615.2**|1599.0|**1228.0**|1225.3|**159.0**|149.7|**54.9**|76.2|**86.1**|89.3|
|(0.65,0.75)|**1616.4**|1596.2|**1226.5**|1218.8|**158.8**|144.9|**61.5**|77.3|**91.9**|95.6|
|(0.54,0.84)|**1613.4**|1598.5|1223.1|**1229.3**|**155.6**|151.2|**58.6**|83.1|**92.3**|97.6|
|(0.41,0.91)|**1612.9**|1594.3|1222.6|**1224.0**|**155.1**|146.5|**57.3**|84.4|**87.3**|93.6|
|(0.28,0.95)|**1613.0**|1596.9|**1230.5**|1218.7|**159.1**|145.1|**61.5**|82.0|**92.4**|95.5|
|(0.14,0.98)|**1612.2**|1591.8|1214.4|**1224.3**|**150.6**|145.4|**58.2**|86.7|**80.7**|90.2|
|(0.00,1.00)|**1613.0**|1592.4|**1228.2**|1224.3|**158.0**|145.7|**63.8**|87.6|**84.5**|87.6|

Second, we varied the truncation thresholds in normal distributions, setting $\omega_0\sim\mathcal{N}_{[-2,2]}(0,1)$. The corresponding results are give in Table 5, which show that our proposed method consistently performs better.

**Table 5**
|JOBS|S-Rewards||L-Rewards||$\Delta{W}$||S-VAR||L-VAR||
|:-:|:-:|:-:|:-:|:-:|:-:|:-:|:-:|:-:|:-:|:-:|
|PreferenceVector|OURS|LW|OURS|LW|OURS|LW|OURS|LW|OURS|LW|
|(1.00,0.00)|1612.9|**1617.4**|**1210.7**|1208.7|147.9|**149.2**|**52.5**|54.4|**101.7**|109.4|
|(0.98,0.17)|**1615.6**|1605.9|1210.3|**1211.1**|**149.1**|144.6|**52.3**|56.1|**95.9**|96.7|
|(0.94,0.34)|**1609.5**|1607.9|**1215.2**|1199.3|**148.5**|139.7|59.1|**55.0**|**104.0**|104.3|
|(0.86,0.50)|**1615.1**|1606.3|1203.8|**1204.7**|**145.6**|141.6|**55.0**|64.1|106.9|**104.6**|
|(0.76,0.64)|**1614.0**|1609.5|1211.4|**1212.1**|**148.8**|146.9|**57.7**|70.8|**91.7**|92.5|
|(0.64,0.76)|**1611.1**|1609.5|**1213.3**|1196.2|**148.3**|139.0|**52.1**|67.5|**100.5**|105.8|
|(0.50,0.87)|**1615.6**|1610.7|**1209.0**|1202.9|**148.4**|143.0|**57.5**|75.1|105.9|**94.5**|
|(0.34,0.94)|**1615.8**|1606.8|**1211.8**|1198.7|**150.0**|138.9|**53.3**|74.1|**96.3**|101.1|
|(0.17,0.98)|**1617.3**|1600.1|**1215.2**|1210.8|**152.4**|141.6|**54.9**|78.3|112.4|**98.1**|
|(0.00,1.00)|**1614.1**|1605.5|1207.7|**1210.0**|**147.0**|143.9|**56.0**|72.2|**94.9**|98.1|


> **W6:** For L-REWARDS in In Table 1, the proposed method is comparable to the LW method.

**Response:** In this article, **we aim to develop a policy learning method that balances both short and long-term rewards, rather than focusing solely on maximizing either of them.**

To evaluate the proposed method, we mainly use three metrics: S-REWARD, L-REWARD, and $\Delta W$. S-REWARD and  L-REWARD measures the short and long-term rewards, and $\Delta W$ measures the balanced welfare, indicating the policy's ability to trade-off between multiple rewards. Clearly, **to evaluate an approach that balances both short and long-term reward, $\Delta W$ is the most important metric (lines 283-284).**

In Table 1, the proposed method may not always perform better on L-REWARD, but it shows superior performance for $\Delta W$. **This indicates that the LW method does not achieve a good balance, as it pursues long-term rewards at the expense of short-term rewards. In contrast, our method proves to be Pareto optimal, pursuing long-term rewards while also maximizing short-term rewards.**


> **W7:** More explanation is needed for the influence of $r$ and $T$ in experiments.

**Response:** The proposed method consists of two parts: (1) estimation of long-term and short-term rewards;(2) learning the optimal policy using DPPL method. In our experimental, different missing rates $r$ and time steps $T$ mainly affect the generation of long-term outcomes, thus impacting the estimation of long-term rewards, i.e., the first part. However, since the proposed DPPL method focuses on the second part. Thus, we may not observe a significant influence of $r$ and $T$ on the results. This indicates the stability of the proposed method.

---
[6] Wu et al. Policy Learning for Balancing Short-Term and Long-Term Rewards. arXiv, 2024

[7] Cheng et al. Long-term effect estimation with surrogate representation, WSDM, 2021.

[8] Li et al. Trustworthy policy learning under the counterfactual no-harm criterion, ICML, 2023.

---

### Author Response · Authors · 2024-08-11
**Gratitude for Review and Invitation for Further Discussion**

Dear Reviewers ouKL, xruw, hVvL, and B23s,

We are deeply grateful to all reviewers who dedicated their valuable time and effort to evaluating our work.

As the rebuttal deadline approaches, we would like to express our appreciation once again and invite any additional questions or concerns you may have. Your insights are crucial to refining our research, and we welcome further discussion.

Sincerely

---

### Decision · Program_Chairs · 2024-09-25

**Decision:**

Accept (poster)

**Comment:**

This paper proposes a method for learning an optimal policy with both long-term and short-term rewards, which may be missing in the causal inference context. The idea is to formulate a multi-objective optimization problem and obtain a set of Pareto optimal policies by solving a series of subproblems based on preference vectors.

Many reviewers were somewhat misled by the title and the abstract into thinking the paper addressed a general long vs. short trajectory setting in reinforcement learning. I suggest the authors update the title and abstract to explicitly state that the main focus of the paper is on policy learning in causal inference. Additionally, as reviewer ``hVvl`` pointed out, I recommend making the subsection on the suboptimality of the linear weighting method self-contained, rather than referring to [1-4], as general RL readers might not be familiar with the topic in causal inference.

Overall, my recommendation is to accept the paper. I suggest the authors incorporate the feedback in the final submission.

[1] Miettinen. Nonlinear Multiobjective Optimization, 2012
[2] Marler et al. The Weighted Sum Method for Multi-Objective Optimization: New Insights, 2010
[3] Censor. Pareto Optimality in Multiobjective Problems, 1977
[4] Ngatchou et al. Pareto Multi-Objective Optimization, 2005